# Voltage-independent GluN2A-type NMDA receptor Ca$^{2+}$ signaling promotes audiogenic seizures, attentional and cognitive deficits in mice

Ilaria Bertocchi [ID] et al.[#]

The NMDA receptor-mediated Ca$^{2+}$ signaling during simultaneous pre- and postsynaptic activity is critically involved in synaptic plasticity and thus has a key role in the nervous system. In *GRIN2*-variant patients alterations of this coincidence detection provoked complex clinical phenotypes, ranging from reduced muscle strength to epileptic seizures and intellectual disability. By using our gene-targeted mouse line (*Grin2a$^{N615S}$*), we show that voltage-independent glutamate-gated signaling of GluN2A-containing NMDA receptors is associated with NMDAR-dependent audiogenic seizures due to hyperexcitable midbrain circuits. In contrast, the NMDAR antagonist MK-801-induced c-Fos expression is reduced in the hippocampus. Likewise, the synchronization of theta- and gamma oscillatory activity is lowered during exploration, demonstrating reduced hippocampal activity. This is associated with exploratory hyperactivity and aberrantly increased and dysregulated levels of attention that can interfere with associative learning, in particular when relevant cues and reward outcomes are disconnected in space and time. Together, our findings provide (i) experimental evidence that the inherent voltage-dependent Ca$^{2+}$ signaling of NMDA receptors is essential for maintaining appropriate responses to sensory stimuli and (ii) a mechanistic explanation for the neurological manifestations seen in the NMDAR-related human disorders with GRIN2 variant-meidiated intellectual disability and focal epilepsy.

---

[#]A list of authors and their affiliations appears at the end of the paper.

N-methyl-D-aspartate receptors (NMDARs) play an essential role in the survival, differentiation, and migration of neurons, as well as in the formation and stabilization of synapses and neuronal circuits both during development and in adulthood[1–5]. The critical role of NMDARs is based on (I) their slow response to the major excitatory neurotransmitter L-glutamate, (II) the voltage-dependent current block by extracellular $Mg^{2+}$, and (III) their high $Ca^{2+}$ permeability[6,7] (for a recent review see ref. [8]). By combining these three features, NMDARs provide a precise and elegant molecular mechanism for the activation of $Ca^{2+}$-dependent postsynaptic second messenger cascades, which trigger specific intracellular responses[9,10]. In turn, these responses are necessary for the experience-dependent priming of neural networks[11].

For several decades the precise coincidence detection of pre- and postsynaptic activity by NMDAR-dependent $Ca^{2+}$ signaling has been postulated to be of crucial importance for learning and adapting to environmental stimuli. However, this has rarely been tested directly at the behavioral level[12,13]. By introducing the well-characterized GluN2A(N615S) mutation (previously called N596)[14,15] into the mouse genome, we were able to study the effects of an inappropriate glutamate-induced $Ca^{2+}$ influx through GluN2A-type NMDARs, even at resting potentials, on synaptic plasticity, activity-induced c-Fos expression, neuronal network activity in the hippocampus and, lastly, on behavior. This analysis had not been possible in previous studies with gene-targeted $Grin2a^{(N614Q)}$ mice that died for unknown reasons 2 weeks after birth[12].

The molecular components responsible for the $Mg^{2+}$-regulated $Ca^{2+}$ influx through the channel are localized at the tip of the ion pore of heterotetrameric NMDARs. The immobile ion pore is assembled from four P-loop structures in the M2 membrane segments of NMDAR subunits: i.e. two obligatory GluN1 subunits and two from the GluN2(A–D) or GluN3(A,B) subunit families (for a review see ref. [16]). Within this complexity, GluN1/2-receptors are the most abundant NMDAR subtypes throughout the central nervous system[17–19]. In these NMDAR subtypes, an asparagine amino acid residue in the GluN1 subunits (N614, labeled previously N598 (ref. [20])) and two neighboring N residues in the GluN2 subunits (N614 and N615, labeled previously N595 and N596 (ref. [15])) located at the tip of the P-loops, build the narrow constriction of the ion channel pore, and differentially modulate $Mg^{2+}$ block and $Ca^{2+}$ permeability[14,15,21].

The amino acid substitutions GluN1(N614Q) and GluN1 (N614R) abolished or reduced both the $Mg^{2+}$ block and $Ca^{2+}$ permeability of all NMDARs. This caused the premature death of the respective mutant mice due to respiratory failure[22], as described in mice completely lacking NMDARs ($Grin1^{-/-}$)[23]. The importance of precise NMDAR signaling for the establishment of autonomic pattern activity in neuronal circuits is further emphasized by $Grin2b$ knockout mice. In GluN2B-deficient pups, the trigeminal neuronal pattern formation is impaired and the pups starve to death within the first days after birth due to the lack of suckling responses[24].

The NMDAR function as a coincidence detector is generally identified with the induction of long-term potentiation (LTP), the dominant experimental model of synaptic plasticity[25]. The voltage-controlled $Mg^{2+}$ block is essential for this activity-dependent NMDAR signaling[10]. In recombinant GluN1/2A NMDARs the $Mg^{2+}$ block is predominantly determined by the asparagine amino acid residue GluN2A(N615). In oocytes and HEK293 cells expressing recombinant GluN1/GluN2A(N615S) heterodimeric receptors, the GluN2A(N615S) mutation led to a pronounced attenuation of the $Mg^{2+}$ block and a 1.4-fold increased $Ca^{2+}$ permeability[14,15]. Notably, a similar $Mg^{2+}$ block attenuating point mutation (c.1841A>G, p.Asn615Ser) at the identical position of the GluN2A subunit was found in two unrelated young female patients who suffered from epileptic seizures, intellectual disability (ID), moderate hypotonia, and speech/language disorders[26,27].

To unravel the functional contributions of the voltage-dependent $Mg^{2+}$ block in neurological disease, brain physiology, and behavior, we generated and analyzed heterozygous and homozygous gene-targeted mice with global $Grin2a^{(N615S)}$ expression ($Grin2a^{+/S}$ and $Grin2a^{S/S}$, respectively). The viability and good health of $Grin2a^{+/S}$ and $Grin2a^{S/S}$ mutant mice allowed us to resolve the functional consequences of this mutation, particularly for seizure susceptibility, hippocampal plasticity, hippocampal oscillatory activity, and in cognitive performance during simple and complex associative learning tasks. Thus, experimental results show that the voltage-dependent $Ca^{2+}$ signaling of GluN2A-type NMDARs is of particular importance for the tight temporal control of attentional processes, which becomes especially important when there are spatial and/or temporal discontiguities between relevant cues and behaviorally relevant outcomes.

## Results

**Generation of GluN2A(N615S)-expressing mice.** Heterologous expression of GluN2A(N615S) (Fig. 1a) with GluN1 demonstrated a reduced $Mg^{2+}$ block of GluN1/2(N615S) receptors in the presence of 1 and 4 mM of $Mg^{2+}$ at hyperpolarized membrane potentials when compared with wild-type NMDARs (Fig. 1b). In the absence of $Mg^{2+}$, short glutamate applications (20 ms) activated mutated and wild-type NMDAR channels with comparable current amplitudes and similar activation (rise time) and deactivation kinetics. During prolonged glutamate applications (600 ms) slower desensitization kinetics were obvious for the GluN1/2A(N615S) compared to GluN1/2A heterodimeric receptors (Fig. 1b and Supplementary Table 1).

By classical gene-targeted replacement[28] we inserted the c.1841A>G mutation at the homologous position in exon 10, and thus replaced the $Grin2a$ asparagine codon (AAT, N615) with a codon for serine (AGT) (Fig. 1c and Supplementary Fig. 1). The $Grin2a$ cDNA sequence analysis of total brain mRNA of heterozygous $Grin2a^{+/S}$ mice together with the comparable GluN2A immunosignals in forebrain extracts of $Grin2a^{+/+}$ and $Grin2a^{S/S}$ littermates verified that adult mice expressed the $Grin2a^{(N615S)}$ and $Grin2a^+$ alleles at the same level (Fig. 1d, e). We also observed in forebrain extracts statistically comparable levels of the GluA1 subunit of the amino-3-hydroxy-5-methyl-4-isoxazolepropionic acid receptor (AMPAR), the postsynaptic marker protein PSD95 and the phosphorylated form of α-CaMKII in $Grin2a^{S/S}$, $Grin2a^{+/s}$, and $Grin2a^{+/+}$ mice. However, GluN2B levels were significantly higher in the membrane fraction but not in the total forebrain fraction from homozygous $Grin2a^{S/S}$ mouse brains when compared to heterozygous and wild-type littermates (Fig. 1e and Supplementary Fig. 2).

**Regular glutamatergic signal transmission but increased GluN2B-type LTP in GluN2A(N615S)-expressing mice.** Patch clamp recordings in CA1 pyramidal cells showed that the GluN2A(N615S) subunits are functionally incorporated in synaptic NMDARs. At −70 mV and in the absence of extracellular $Mg^{2+}$, the synaptic AMPA/NMDA current ratio and the individual peak of AMPAR and NMDAR currents were similar in CA1 cells from $Grin2a^{S/S}$, $Grin2a^{+/S}$, and $Grin2a^{+/+}$ mice (Fig. 2a, left). However, the presence of GluN2A(N615S) in synaptic NMDAR was indicated by the decreased AMPAR/NMDAR response ratio detected in $Grin2a^{S/S}$ mutants in the presence of 1 mM $Mg^{2+}$ (Fig. 2a, right). This NMDAR-induced current increase was more pronounced at postnatal day 42 (P42)

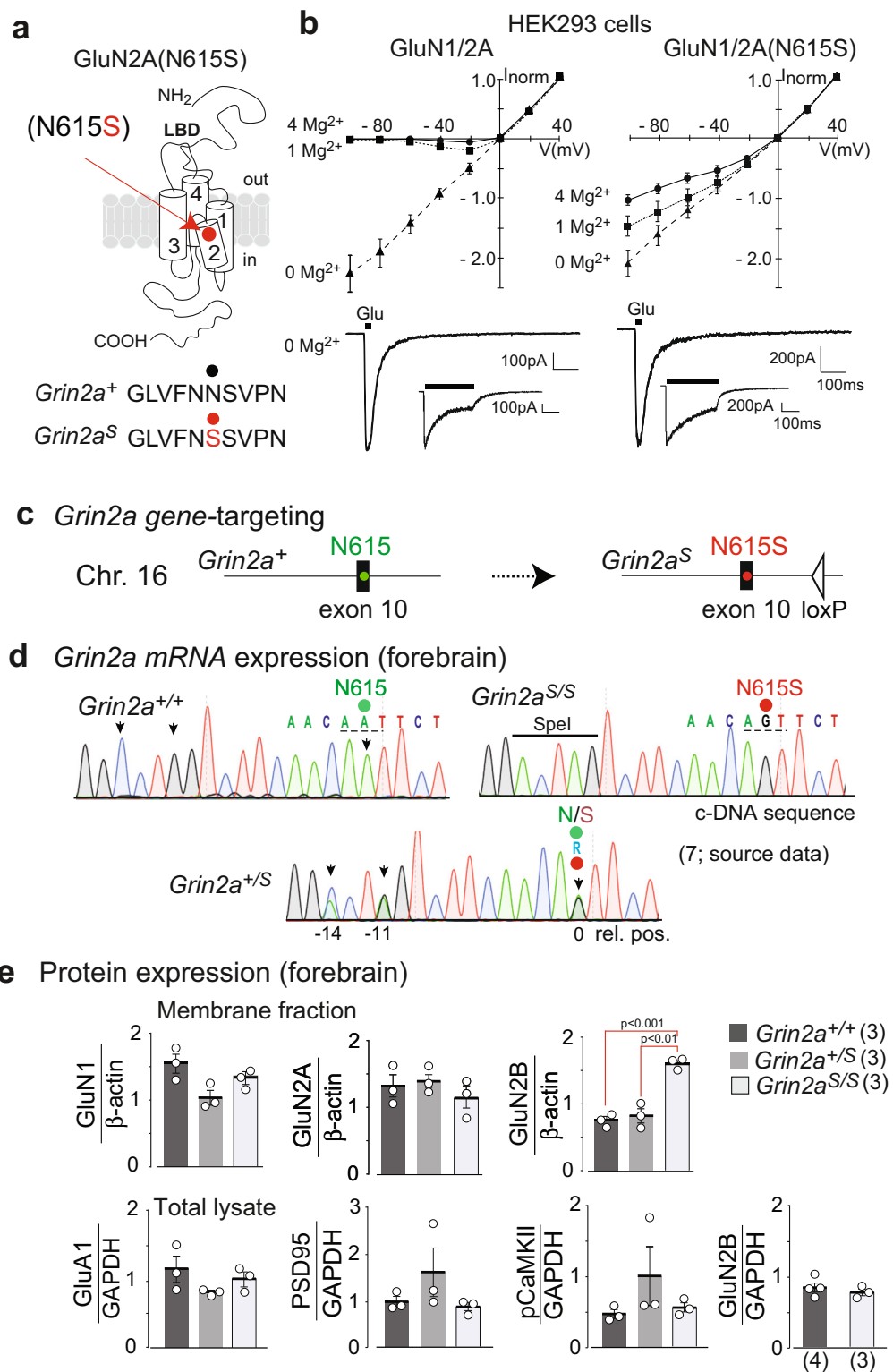

compared to postnatal day 14 (P14), in accordance with the increasing developmental *Grin2a* expression profile after birth[18,29] (Fig. 2a, right). Thus, the homozygous *Grin2a^S/S* mice express significant amounts of synaptic GluN2A(N615S)-containing NMDARs with reduced voltage dependence.

To directly assess changes in excitatory synaptic transmission and synaptic excitability, we recorded simultaneously in the apical dendritic and soma layers in the CA1 region of hippocampal slices from GluN2A(N615S)-expressing mice and wild-type littermates. First, we measured the fiber volley, the field excitatory postsynaptic potential (fEPSP), and the population spike as a function of different stimulation strengths. In our field recordings the stimulation strength required to induce pre-volley amplitude of 1.0 or 1.5 mV was statistically unaltered in GluN2A(N615S)-

**Fig. 1 GluN2A(N615S) containing NMDAR expression in vitro and in vivo. a** The position of N615 in the membrane segment M2 is depicted together with the three channel-forming trans-membrane segments M1, 3, and 4. **b** In HEK293 cells, recombinantly expressed GluN1/GluN2A and GluN1/GluN2A (N615S) channels were activated by fast glutamate application (1 mM; in the continuous presence of the co-agonist glycine, 10 µM) at holding potentials from –100 to +40 mV in different extracellular $Mg^{2+}$ concentrations. NMDAR-mediated peak currents were normalized to those obtained at +40 mV. Data points represent mean ± SEM for $n = 4$–7 different HEK293 cells. Representative current traces evoked in 0 mM $Mg^{2+}$ at – 60 mV, with 20 and 600 ms applications, are shown below the *IV* plots and were used to determine the current kinetics (Supplementary Table 1). **c** Schematic view of the A to G replacement in exon 10 of the mouse *Grin2a* gene. **d** Reverse transcription PCR (RT-PCR)-sequence analyses of total brain mRNA show the A-to-G mutation in pos. 0 and two diagnostic silent mutations at pos. –11 and –14 in the pore loop encoding gene segment in $Grin2a^{+/+}$, $Grin2a^{S/S}$, $Grin2a^{+/S}$ mice. In $Grin2a^{+/S}$ mice, the overlay of two different colored "nucleotide" peaks, at position 0, –11, and –14 indicate equimolar amounts of mRNA from the $Grin2a^+$ and the targeted $Grin2a^S$ alleles. **e** Immunoblots of forebrain protein lysates of 4-week-old mice (Supplementary Fig. 2) indicate no genotype-specific differences of GluN1, GluN2A, and the AMPAR subunit GluA1 expression relative to the β-actin levels ($p > 0.05$). The levels of PSD95 and αCaMKII (in its phosphorylated state, pCaMKII) are also comparable between genotypes relative to the GAPDH expression. The GluN2B expression level in the membrane fraction was significantly increased in $Grin2a^{S/S}$ mice when compared to $Grin2a^{+/+}$ and $Grin2a^{+/S}$ mice but not in the levels of total protein lysates (for statistics: Supplementary Statistics to Fig. 1). The number of mice is given in brackets. Error bars represent mean ± SEM.

expressing mice and showed only a trend towards lower fEPSP amplitudes at a given pre-volley amplitudes of 1.5 mv in $Grin2a^{S/S}$ mice. Together with the similar paired-pulse ratio our field recordings revealed no major alterations in CA3-to-CA1 synaptic transmission of $Grin2a^{S/S}$ and $Grin2a^{+/S}$ mice (Fig. 2b).

To analyze whether these voltage-independent GluN1/2A (N615S) receptors can still induce synaptic plasticity, we analyzed field LTP (fLTP) at CA3-to-CA1 synapses in the different *Grin2a* genotypes. Here we found that the magnitude of hippocampal fLTP in $Grin2a^{S/S}$ and $Grin2a^{+/S}$ mice was unaffected ex vivo and in vivo (Fig. 2c, d[30]), in contrast to the reduced fLTP found in GluN2A-deficient mice and in mice lacking the GluN2A intracellular C-terminal domain of the GluN2A subunit[31,32]. This suggests that the coincidence signaling of GluN1/2(N615S) receptors is still operative. However, since the GluN2B antagonist CP101,106 significantly reduced the fLTP in $Grin2a^{S/S}$ and $Grin2a^{+/S}$ mice but not in $Grin2a^{+/+}$ littermates (Fig. 2e), we conclude that (i) pure GluN1/2A(N615S) receptors have a reduced contribution to the long-term synaptic enhancement after tetanic stimulation and (ii) the fLTP recorded in $Grin2a^{S/S}$ and $Grin2a^{+/S}$ is substantially mediated by GluN2B-containing receptors.

This conclusion was strengthened by using four tetanic stimulations ($4 \times 100$ Hz), which can induce GluN2B-dependent LTP in the absence of functional GluN2A[33,34]. In comparison with single tetanic stimulation, LTP was significantly increased 40–45 min after the $4 \times 100$ Hz stimulation in hippocampal slices of both $Grin2a^{S/S}$ and $Grin2a^{+/S}$ mice compared to WT control littermates. This LTP increase was reduced by CP101,606 (Supplementary Fig. 3a), an effect that is reminiscent of the one described for LTP reduction in juvenile (P14) wild-type mice[35]. This LTP was still completely NMDAR-dependent and could be blocked by the NMDAR antagonist APV (Supplementary Fig. 3b). Together, these results show the incorporation of GluN2A (N615S) into synaptic NMDARs but reduced contribution of GluN2A(N615S) receptors in LTP.

**GluN2A(N615S) homozygous mice show altered home cage behaviors but regular brain anatomy, no apoptosis or neurodegeneration.** In contrast to other genetically modified mice with altered NMDAR $Ca^{2+}$ permeability and/or altered $Mg^{2+}$ block[12,22], we found that $Grin2a^{S/S}$ and $Grin2a^{+/S}$ mice are viable and long-living. However, $Grin2a^{S/S}$ mice can be recognized by their reduced body weight. Moreover, $Grin2a^{S/S}$ mice showed poor nest building and burrowing activities (Fig. 3a), which can be indicators of impairments associated with hippocampal dysfunction[36–38]. $Grin2a^{S/S}$ mice also exhibited the paw- and limb-clasping reflex (Fig. 3b and Supplementary Data: Video 1)[39–41], reduced muscle strength, lower activity in the running wheel (Fig. 3c and

Supplementary Fig. 4a), and decreased grip strength and climbing activity (Supplementary Fig. 4b–d). In marked contrast, the general locomotor activity of $Grin2a^{S/S}$ mice that we recorded automatically in the LABORAS home cage[42] was significantly increased in the first 5 h of the night cycle (Supplementary Fig. 4d), although there was no difference in the maximum and mean running speed between genotypes. The number of rearing, grooming, and eating events was not affected (Supplementary Fig. 4d). Our analysis in the Catwalk test showed that $Grin2a^{S/S}$ mice exhibit a regular walking pattern (Supplementary Fig. 4e) and can achieve normal balance scores in the stationary rod test (Supplementary Fig. 4f). Some minor alterations in the base support of the hind limbs (the distance of hind limbs during walking) of $Grin2a^{S/S}$ mice (Supplementary Fig. 4e) might contribute to the slightly delayed acquisition in the rotarod test (Supplementary Fig. 4g). In heterozygous $Grin2a^{+/S}$ mice, we found a trend towards reduced activity in the running wheel and climbing (Supplementary Fig. 4a, d), and 20–30% of $Grin2a^{+/S}$ mice did show strong paw- and limb-clasping.

The reduced $Mg^{2+}$ block of GluN1/2A(N615S) receptors could conceivably permit a glutamate-induced, voltage-uncontrolled $Ca^{2+}$ influx into neurons that are sensitive to $Ca^{2+}$-induced toxicity. However, we did not detect any signs of cytotoxicity or neurodegeneration in Nissl-stained brain slices from adult $Grin2a^{S/S}$ mice (Fig. 3d). Furthermore, no chromosomal DNA degradation could be detected in the terminal deoxynucleotidyl transferase dUTP nick end labeling (TUNEL) assay in the hippocampus and adjacent cortical cell layers, suggesting that there was no apoptosis or necrosis in $Grin2a^{S/S}$ brains (Fig. 3e). Our Timm staining of mossy fibers did not indicate hippocampal sclerosis, a neuropathological marker of temporal lobe epilepsy in humans and rodents[32,43,44], not even in 6-months-old $Grin2a^{S/S}$ mice (Fig. 3f). Lastly, the immunosignals of neuronal and astrocytic markers NeuN and GFAP were comparable between controls and $Grin2a^{S/S}$ mice, as was the hippocampal layer-specific distribution of calbindin, the interneuronal protein parvalbumin, and the AMPAR subunit GluA1 (Fig. 3g).

***Grin2a^{S/S}* mutant mice are highly sensitive to audiogenic seizures (AGSs).** Considering the presence of an epileptic phenotype in the patient with the analogous GluN2A(N615K) mutation[27] and the transient epileptiform discharges observed in GluN2A-deficient mice[45], we analyzed the seizure susceptibility of GluN2A (N615S)-expressing mice. When exposed to a high-frequency acoustic stimulus (11 kHz), which is used for AGS induction in DBA mice[46], all $Grin2a^{S/S}$ mutants responded immediately after tone onset with a stereotypic AGS response composed of wild running followed by clonic seizures, tonic extension of limb extremities, and respiratory arrest (RA). In contrast, no seizures were observed for $Grin2a^{+/+}$ control mice, whereas in $Grin2a^{+/S}$

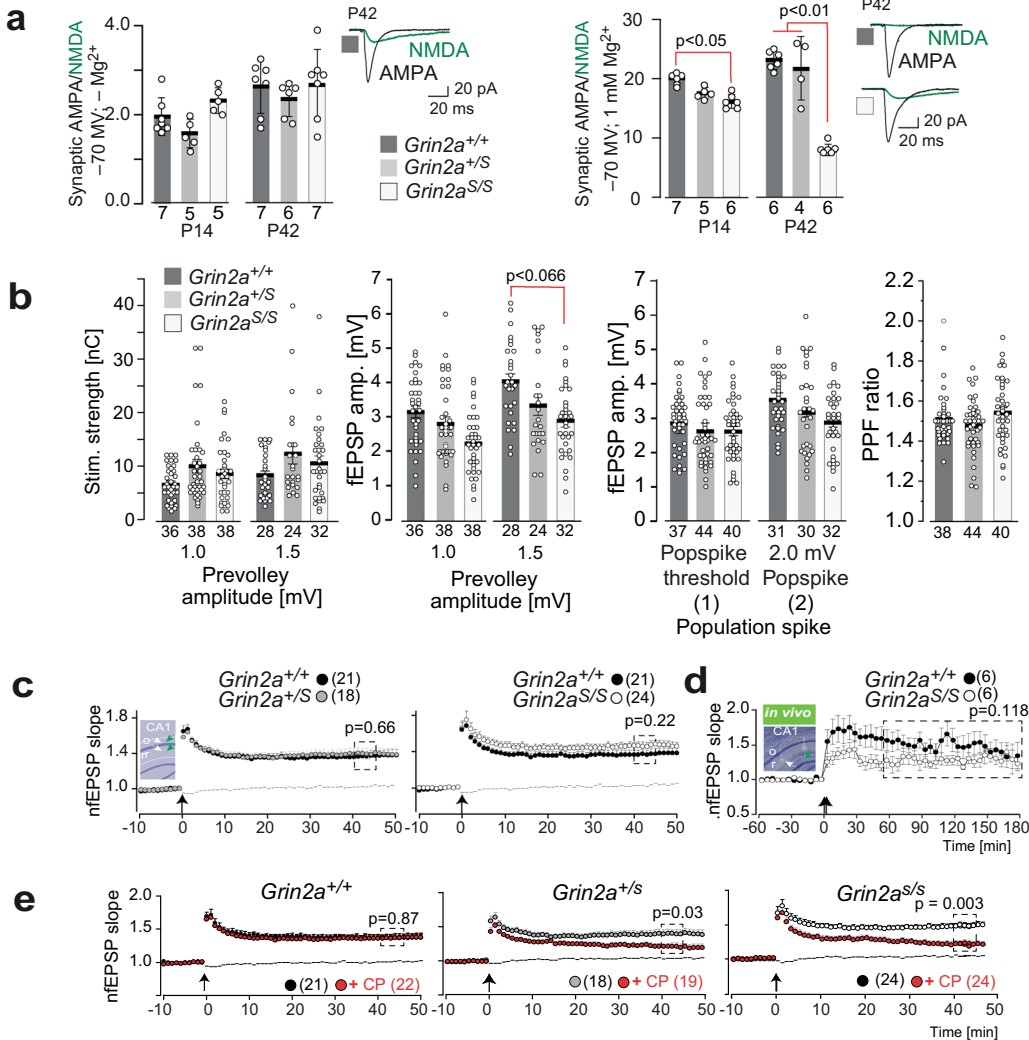

**Fig. 2 Hippocampal synaptic transmission and plasticity in *Grin2a^{S/S}* and *Grin2a^{+/S}* mice with the GluN2A(N615S) mutation. a** (left) In the absence of extracellular Mg$^{2+}$ the synaptic AMPA/NMDA ratio of CA1 pyramidal cells in acute hippocampal slices is not altered in *Grin2a^{+/S}* and *Grin2a^{S/S}* mice compared to control littermates. (right) In the presence of extracellular Mg$^{2+}$ the strong reduction of the AMPA/NMDA ratio in *Grin2a^{S/S}* mice relates to the increased NMDA currents at CA1 synapses (Tukey's test). Example traces are depicted to the right of each bar graph. Data from the same experimental group were pooled across animals and are presented as mean ± SEM (see also ref. [35]) with $p < 0.05$ being designated as statistically significant. Numbers in bar graphs indicate the number of slices. **b** Paired-pulse facilitation at CA3-to-CA1 synapses excluded strong alterations of presynaptic function in mutant mice. The stimulation strengths (in nC) necessary to elicit a pre-volley of 1.0 and 1.5 mV and the resulting fEPSP amplitudes were comparable in all genotypes but showed only a trend towards lower fEPSP amplitudes recorded at 1.5 mV pre-volley amplitudes in *Grin2a^{S/S}* mice. The fEPSP amplitudes necessary to elicit a just detectable population spike (1) and a population spike of 2 mV amplitude (2) and the paired-pulse facilitation ratio (PPF) at an interstimulus interval of 50 ms did not indicate any synaptic impairments in *Grin2a^{S/S}* and *Grin2a^{+/S}* mice compared to WT littermates. The number of slices is indicated in the bar graphs or in brackets. **c** Field LTP (fLTP) at CA3-to-CA1 synapses, induced by tetanic stimulation in slices (1 s; 100 Hz; arrow), was comparable in all three genotypes. The inset in **c** gives a schematic view of the stimulating (white arrow) and the recording (green arrow) electrode positions in str. radiatum (r) and str. oriens (o). **d** Similarly, CA1-to-CA3 fLTP (induction: 2 × 1 s; 100 Hz; arrow) in freely moving mice was comparable between *Grin2a^{S/S}* and *Grin2a^{+/+}* mice. The inset shows a Nissl-stained slice of one recorded mouse post mortem. **e** In mutant mice, but not in control littermates, the GluN2B-containing NMDAR contributes significantly to the magnitude of LTP, since LTP was significantly reduced by the GluN2B-specific antagonist CP101,106 (CP). Recordings of the non-tetanized control pathway in **c** and **e** are given as dashed lines. Error bars represent mean ± SEM (for statistics: Supplementary Statistics to Fig. 2).

mice the audiogenic stimulus had lower penetrance and in one of the five epileptic heterozygotes (out of a total of 17 *Grin2a^{+/S}* mice tested), the seizures were not followed by RA (Fig. 4a and Supplementary Data: Video 2). In *Grin2a^{S/S}* mice, death from RA could be prevented by chest massage within a few seconds after the tonic phase, allowing us to visualize in revived *Grin2a^{S/S}* mice 90 min post-AGS a pronounced elevation of the immediate early genes c-Fos and ARC, specifically in the ventromedial nucleus of

the hypothalamus, the medial amygdala, and in the inferior colliculus (IC) and periaqueductal gray (PAG) midbrain nuclei (Fig. 4b and Supplementary Fig. 5a). The very same AGS-specific activity network, which excludes a major hippocampal contribution, has been described in detail in other kindling-based animal models for AGS[47,48]. Indeed, the hippocampus of *Grin2a^{S/S}* mice was spared from the AGS-induced increase in c-Fos and ARC expression (Fig. 4b and Supplementary Fig. 5a).

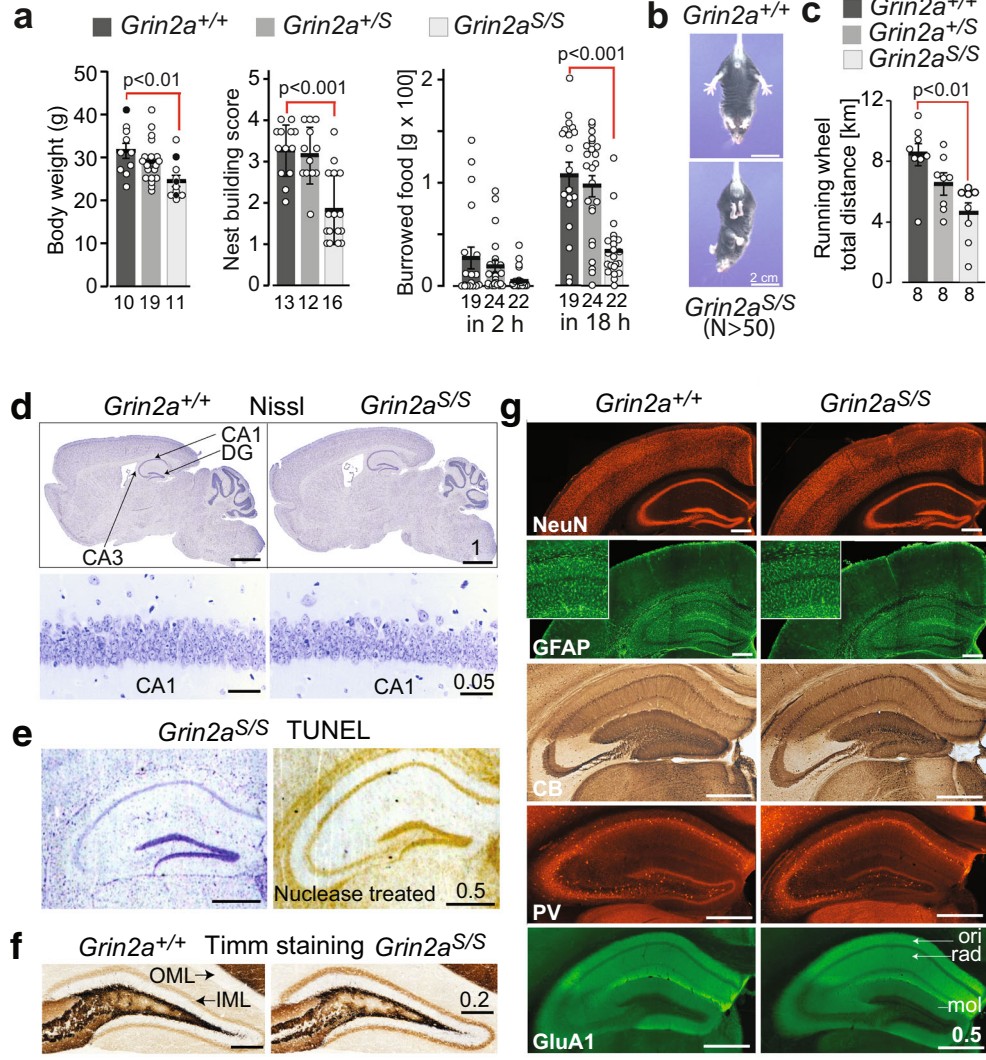

**Fig. 3 Grin2a$^{S/S}$ mice are viable and show no signs of neurodegeneration or altered brain structure. a** Grin2a$^{S/S}$ mice had a reduced body weight (by 18% in adults), significantly lower score in nesting and in overnight burrowing activity (filled circles: 30-week-old mice, white dots: 18-week-old mice). **b** Grin2a$^{S/S}$ mice showed the clasping reflex. **c** During the 12 h dark phase on the Lafayette running wheel the running distance is reduced in Grin2a$^{S/S}$ mice. **d** Nissl staining revealed unaltered cell density throughout the brain (top) and layering of hippocampal subfields, including CA1 (bottom), in Grin2a$^{S/S}$ mice compared to Grin2a$^{+/+}$. Error bars represent mean ± SEM. **e** In the TUNEL assay no apoptotic cells could be detected in the hippocampus of Grin2a$^{S/S}$ mice, compared to the staining of the nuclease-treated positive control slice. **f** No aberrant mossy fiber sprouting in either genotype in the dentate gyrus inner (IML) and outer molecular layer (OML) can be found by Timm staining of coronal sections. **g** The distribution of NeuN-positive neurons and of GFAP-positive glial cells are indistinguishable between Grin2a$^{S/S}$ and Grin2a$^{+/+}$ mice. Mossy fiber projections visualized by anti-calbindin (CB) staining, as well as numbers of parvalbumin (PV)-positive interneurons are similar between both genotypes. Anti-GluA1 immunosignal in all hippocampal layers is comparable between brain sections of Grin2a$^{S/S}$ mice and control littermates. CA1 cornu ammonis region 1, CA3 cornu ammonis region 3, DG dentate gyrus, mol stratum moleculare, ori stratum oriens, rad, stratum radiatum. Scale bars in **d**–**g** are in mm. The number of animals is given below the bars. For the Nissl stain, Tunnel test and Timm stain 3 mice were used per genotype. For the immunohistological analysis of glial fibrillary acidic protein (GFAP), neuronal nuclear antigen (NeuN), Calbindin (CB), and Parvalbumin (PV) five mice and for the GluA1 immunofluorescence stain, three mice were used (for statistics: Supplementary Statistics to Fig. 3).

Together, our data show that increased neuronal excitatory activity in brainstem, midbrain, and extra-hippocampal forebrain regions is associated with AGS in mice with voltage-uncontrolled Ca$^{2+}$ signaling via GluN2A(N615S)-containing NMDARs.

**AGSs in Grin2a$^{S/S}$ mice are rescued by pre-treatment with NMDAR antagonists.** In several animal models, AGS can be prevented by pre-treatment with NMDAR antagonists (for a recent review see ref. [49]). Accordingly, a single i.p. injection of memantine (5 mg/kg) was sufficient to abolish AGS in our Grin2a$^{S/S}$ mutants. The AGS-RA rate dropped from 100% in vehicle-treated Grin2a$^{S/S}$ mice down to 0% in memantine-injected mice, when tested 3 h after injection (Fig. 4c). The typical AGS-induced c-Fos expression could not be detected in brains of memantine-treated, AGS-resistant Grin2a$^{S/S}$ mice (Fig. 4d and Supplementary Fig. 5b). The audiogenic stimulation was then repeated on the same memantine-treated mice on subsequent days. The pharmacological blockade of AGS lasted up to 3 days in some Grin2a$^{S/S}$ mice (three out of six mice; Fig. 4c). When we tested

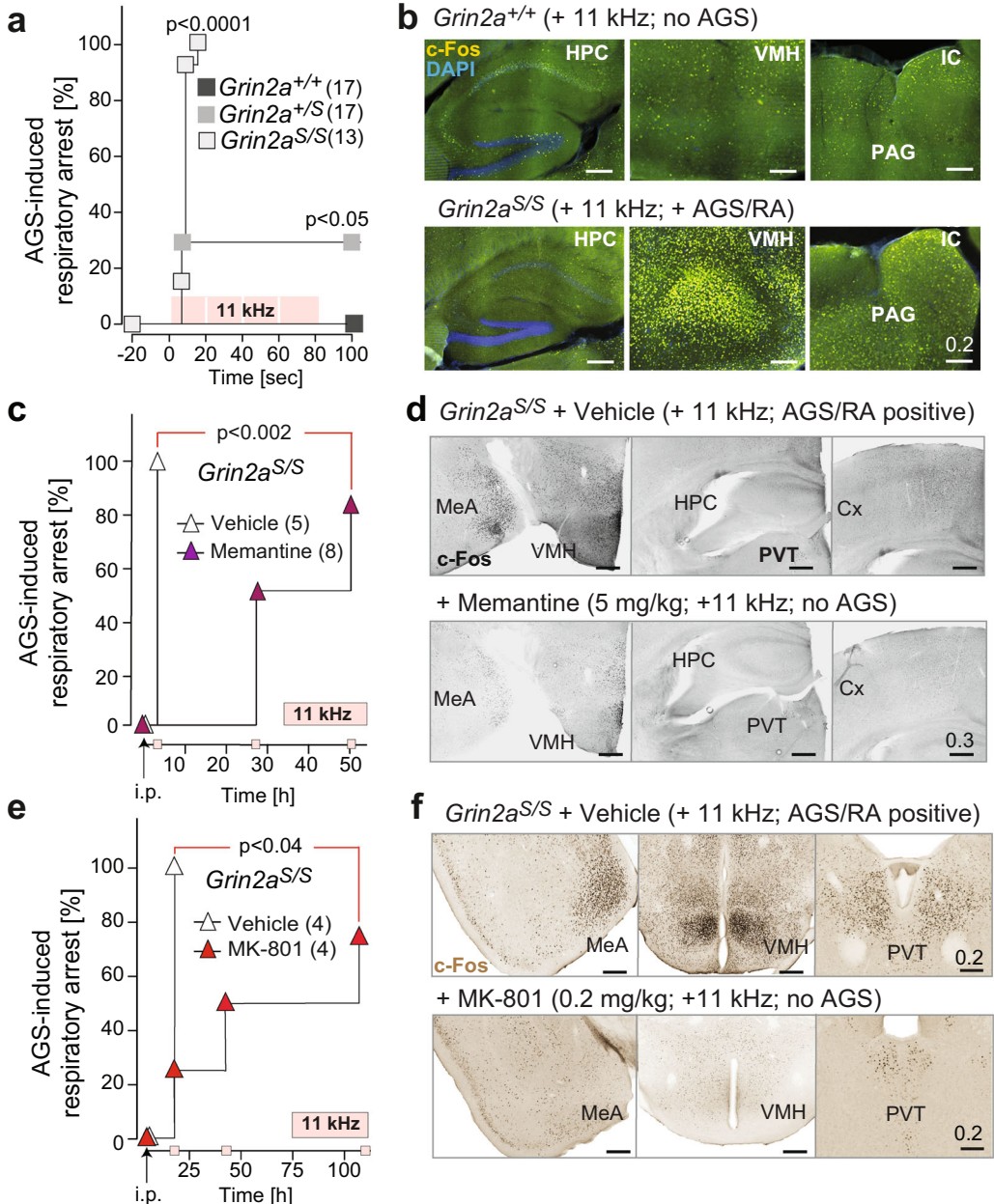

**Fig. 4 $Grin2a^{S/S}$ mice are susceptible to brainstem-derived AGS that can be rescued by NMDAR antagonists. a** Mortality curve showing that none of the $Grin2a^{+/+}$ controls were affected, while audiogenic seizures (AGSs) followed by respiratory arrest (RA) were induced in all $Grin2a^{S/S}$ mice and in a subset of heterozygous $Grin2a^{+/S}$ mutants during the 11 kHz tone exposure [4 repetitions × 20 s tone, 2 s brake; pink squares; $p < 0.0001$ by Log-rank (Mantel–Cox) test]. **b** The c-Fos immunoreactivity was specifically increased in the hypothalamus (VMH), the inferior colliculus (IC), and periaqueductal gray (PAG) but not in the hippocampus (HPC) of a resuscitated $Grin2a^{S/S}$ mouse 90 min after AGS when compared to tone-exposed $Grin2a^{+/+}$ littermates. **c** Memantine i.p. injection in $Grin2a^{S/S}$ mice, 3 h before tone exposure, rescued AGS susceptibility in $Grin2a^{S/S}$ mice. The rescue effect could still be observed in four out of eight, and in two out of eight, mice 27 and 51 h after memantine treatment, respectively. **d** Increased c-Fos immunofluorescence (in grayscale) in the medial amygdala (MeA) and the VMH of a $Grin2a^{S/S}$ animal with AGS (saline injection) compared to a memantine rescued $Grin2a^{S/S}$ littermate. In the paraventricular nucleus of the thalamus (PVT), the hippocampus (HPC) and cortex (Cx) there was no difference in c-Fos expression between memantine-injected and saline-injected animals (for fluorescence images, see Supplementary Fig. 5). **e** As in Fig. 4c but now MK-801 and saline are used for pre-treatment. The MK-801 effect was longer lasting compared to memantine and one out of four animals showed resistance to tone exposure even 4 days after MK-801 injection. **f** Decreased c-Fos DAB immunosignal in an MK-801 AGS-rescued $Grin2a^{S/S}$ animal (bottom) in the medial amygdala (MeA), the VMH and the PVT when compared to saline-injected $Grin2a^{S/S}$ littermate (top). Scale bars are in mm (for statistics: Supplementary Statistics to Fig. 4).

with an alternative NMDAR blocker, MK-801, at low doses (0.2 mg/kg i.p.), the AGS inhibition was similarly effective and even longer lasting (Fig. 4e). The efficiency of MK-801 in reducing the incidence of AGS was highly significant 18 h post-injection and lasted up to 4 days in half of the MK-801-treated $Grin2a^{S/S}$ mice

($N = 4$). Again, AGS-resistant, MK-801-injected $Grin2a^{S/S}$ mice showed no c-Fos induction in those brain regions normally associated with AGSs (Fig. 4f). The pharmacological rescue of AGS in GluN2A(N615S)-expressing mice points to a direct involvement of the aberrant NMDAR signaling in the exaggerated

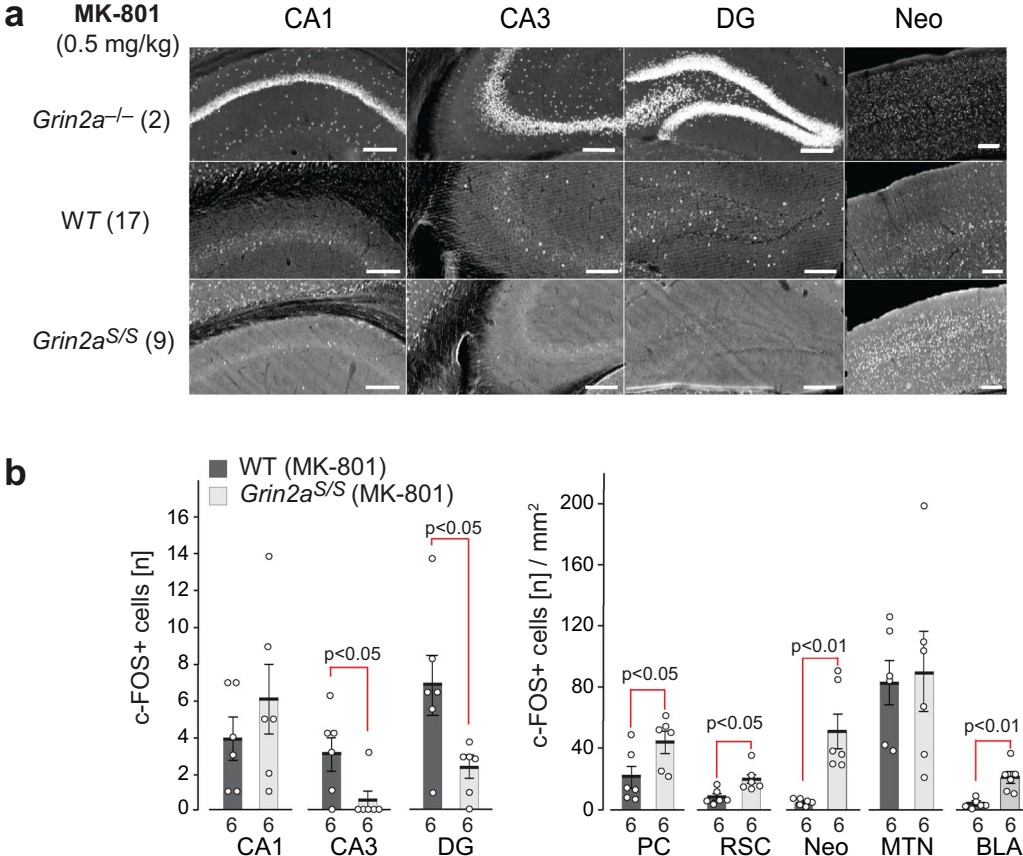

**Fig. 5 MK-801-induced c-Fos expression is reduced in DG granular and CA3 pyramidal cells of *Grin2a^{S/S}* mice.** C-Fos expression 120 min after i.p. injection of MK-801. **a** C-Fos DAB immunosignals are depicted for one representative brain section of the different genotypes as indicated. The original images, including the cortical regions and results of PBS injected animals, are presented in Supplementary Fig. 6. The number of mice analyzed is given in brackets. **b** For quantitative comparisons the total numbers of c-Fos-positive cells in the cellular layers of the hippocampus were counted. C-Fos-positive cells in extra-hippocampal regions are given per mm². Slices from six mice per genotype were used for the statistical analyses (two-tailed *t*-tests) and error bars represent mean ± SEM. CA1 and CA3 cornu ammonis regions 1 and 3, DG dentate gyrus, PC piriform cortex, RSC retrosplenial cortex, Neo neocortex, MTN midline thalamic nuclei, BLA basolateral amygdala. All scale bars: 0.2 mm.

pathological response to a stressful, acoustic stimulus. This result is also in accordance with the resistance of GluN2A-deficient mice to audiogenic-like seizures induced by electrical stimulation of the principal midbrain nucleus, the IC[50].

**Altered drug-induced c-Fos expression in brains of *Grin2a^{S/S}* mice.** Next, we wondered whether alterations in excitatory activity could also be observed in the hippocampus of *Grin2a^{S/S}* mice. As we have previously described[51,52], *Grin2a^{-/-}* mice displayed a robust and significantly increased excitation in hippocampal neurons—visualized by the number of c-Fos-expressing neurons 60–90 min post-MK-801 treatment compared to wild-type controls (Fig. 5a). Now we found in neurons of *Grin2a^{S/S}* mice exactly the opposite: a reduced excitation. We observed only sparsely distributed c-Fos immunoreactive cells in all hippocampal subfields in *Grin2a^{S/S}* mice, and their number was even lower in both CA3 and DG when compared to MK-801-treated wild-type mice (Fig. 5a, b). However, in several extra-hippocampal regions, like the piriform cortex, the retrosplenial cortex, the neocortex, and the basolateral amygdala, but not in the midline thalamic nuclei, c-Fos expression was increased after MK-801 injection in *Grin2a^{S/S}* mice compared to WT controls (Fig. 5b and Supplementary Fig. 6a, b). Since the NMDAR antagonists MK-801 and memantine reside within the channel

vestibule, snuggling into the binding pocket[53], and memantine, as well as the MK-801 blockade of NMDARs, is reduced in recombinantly expressed NMDARs that carry mutations at analogous positions in the pore loop (e.g. GluN2A(N615K) GluN2A (N615Q) and GluN2B(N615Q))[54,55], it seems most likely that blockade of activity-responding GluN1/2B receptors is responsible for the genotype-specific MK-801-induced c-Fos expression. The MK-801-mediated inhibition of GluN1/2B activity of GABAergic interneurons has been proposed as a causal mechanism for the disinhibition of CA1 neurons in GluN2A-deficient mice[56]. Thus and in turn, the constant glutamate-triggered GluN1/2A(N615S) Ca²⁺ signaling may promote the inhibition of principal cells in the hippocampal pathways.

**GluN2A(N615S) expression impairs synchronization of hippocampal activity.** To provide additional evidence for altered hippocampal activity in GluN2A(N615S)-expressing mice, we next analyzed neuronal activity synchronization, which is known to depend critically on NMDAR signaling[57–60]. Our in vivo recordings of hippocampal local field potentials in freely moving animals revealed that the average theta range was similar for wild-type and *Grin2a^{+/S}* mice —at 8.61 ± 0.07 and 8.75 ± 0.28 Hz, respectively— whereas in *Grin2a^{S/S}* mice the theta range peak frequency was reduced to 7.43 ± 0.24 Hz. Interestingly, during

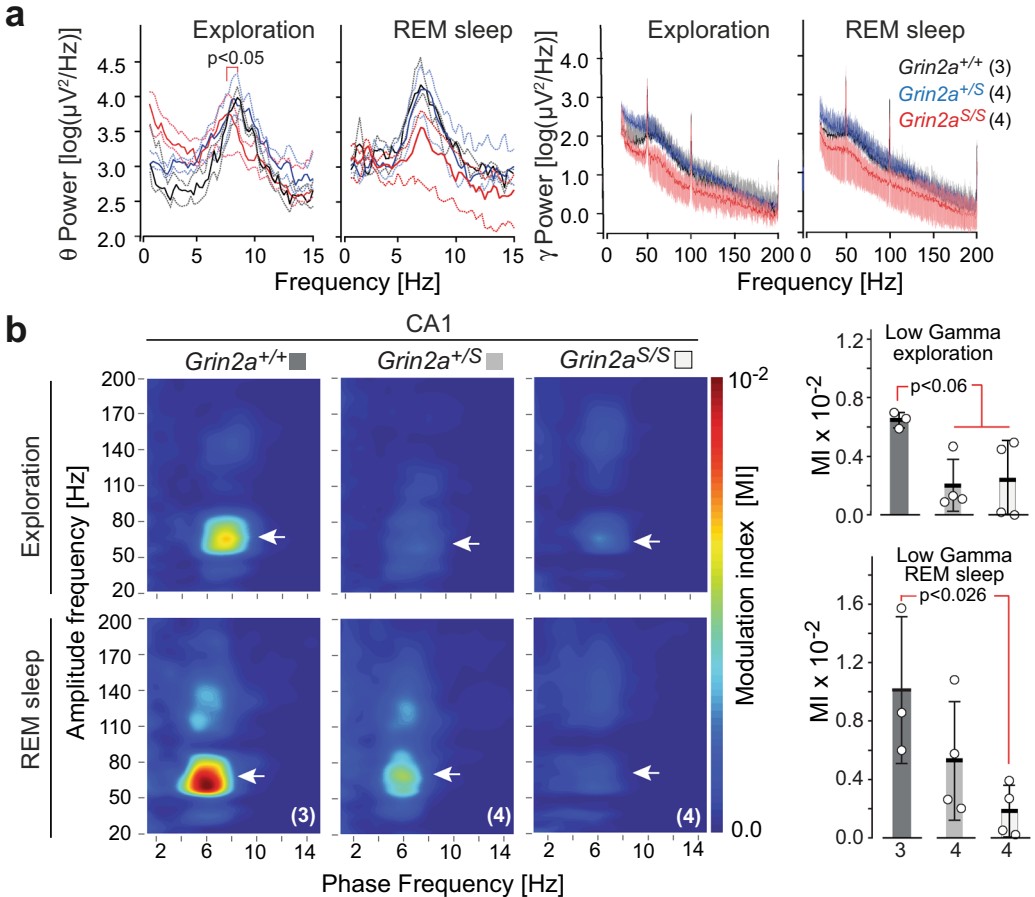

**Fig. 6 Adult GluN2A(N615S) mutant mice express altered hippocampal oscillations during wakefulness and REM sleep. a** (left) Peak frequencies during exploration and rapid eye movement (REM) sleep for theta (0–15 Hz, left diagrams) and gamma oscillations (15–200 Hz, right diagrams) in all genotypes. During exploration, the average theta range peak frequencies were at 8.61 ± 0.07 and 8.75 ± 0.28 Hz for *Grin2a*[+/+] mice and *Grin2a*[+/S] mice, respectively. In *Grin2a*[S/S] mice the theta range peak frequency was significantly slower, measuring 7.43 ± 0.24 Hz (Kruskal Wallis test, $p = 0.0159$). During REM sleep the average frequency ranges of *Grin2a*[+/+] and *Grin2a*[+/S] mice were around 7.00 ± 0.20 Hz whereas *Grin2a*[S/S] mice had a lower but not significantly different range 6.71 ± 0.22 Hz. **a** (right) No significant differences on the low and high gamma frequency bands—ranges between 20–100 Hz and 100–200 Hz, respectively[111]—could be detected between genotypes, neither during exploration nor in REM sleep. The number of mice used in this experiment are given with the name of the genotypes. **b** (left) *Grin2a*[S/S] and *Grin2a*[+/S] mice show reduced theta-gamma phase–amplitude coupling as determined by the modulation index (MI) in the low gamma component in CA1 (white arrowheads) during awake and REM sleep states. **b** (right) For *Grin2a*[S/S] the MI value dropped significantly during exploration and REM sleep although the MI reduction reached significance only during exploration (one-way ANOVA, Kruskal–Wallis test). Numbers below bars of bar graphs indicate the number of animals. Error bars represent mean ± SEM.

rapid eye movement (REM) sleep we observed no significant differences between the three genotypes (Fig. 6a). Thus, the leaky $Mg^{2+}$ block of GluN2A(N615S)-containing NMDARs in *Grin2a*[S/S] mice has a significant effect on theta oscillations during exploratory behaviors but not during REM sleep, suggesting that this effect may be due to differences related to specific behavioral state-dependent activity[61,62].

A further impairment of hippocampal oscillations in GluN2A (N615S)-expressing mice became apparent when we analyzed the coupling between the phase and the amplitude of slow and fast oscillations, as measured by the modulation index (MI)[63–66]. *Grin2a*[+/+] controls showed the typical phase–amplitude coupling patterns of hippocampal networks, with MI values varying between 0.006 and 0.007 in awake animals and 0.006 and 0.016 in REM sleeping mice[67,68] (Fig. 6b). Compared to control animals, the MI values for *Grin2a*[S/S] and *Grin2a*[+/S] animals dropped down significantly to 0.0001–0.005 during exploration. In REM sleep the MI reduction was significant only for *Grin2a*[S/S] mice, dropping down to the range 0.0002–0.004, confirming a robust disruption of complex hippocampal oscillation patterns.

***Grin2a*[S/S] mice are hyperactive and exhibit abnormal and dysregulated attention**. To comprehend the impact of these physiological alterations on the behavioral phenotype of GluN2A (N615S)-expressing mice, we used a battery of different behavioral tests. In accordance with the novelty-induced hyperactivity in the LABORAS cage (Supplementary Fig. 4d), *Grin2a*[S/S] mice also displayed a pronounced locomotor hyperactivity in another novel environment test, without any sign of habituation throughout the entire 2 h session (Fig. 7a). We observed the same increase in exploratory behavior when we exposed *Grin2a*[S/S] mice to five novel objects in three repeated, 6 min sessions in a novel arena. *Grin2a*[S/S] mutants did not decrease their object exploration during the three sessions, in contrast to what was observed for both *Grin2a*[+/S] and *Grin2a*[+/+] control mice (Fig. 7b).

Hyperactivity is often associated with impulsive behavior that can be measured in the cliff avoidance reaction (CAR) test[69]. Here we found that the latency to the first fall from the platform was significantly shorter in both *Grin2a*[+/S] and *Grin2a*[S/S] mice compared to wild-type littermates (Fig. 7c). In addition, during the entire test, the total number of falls for *Grin2a*[S/S] animals was

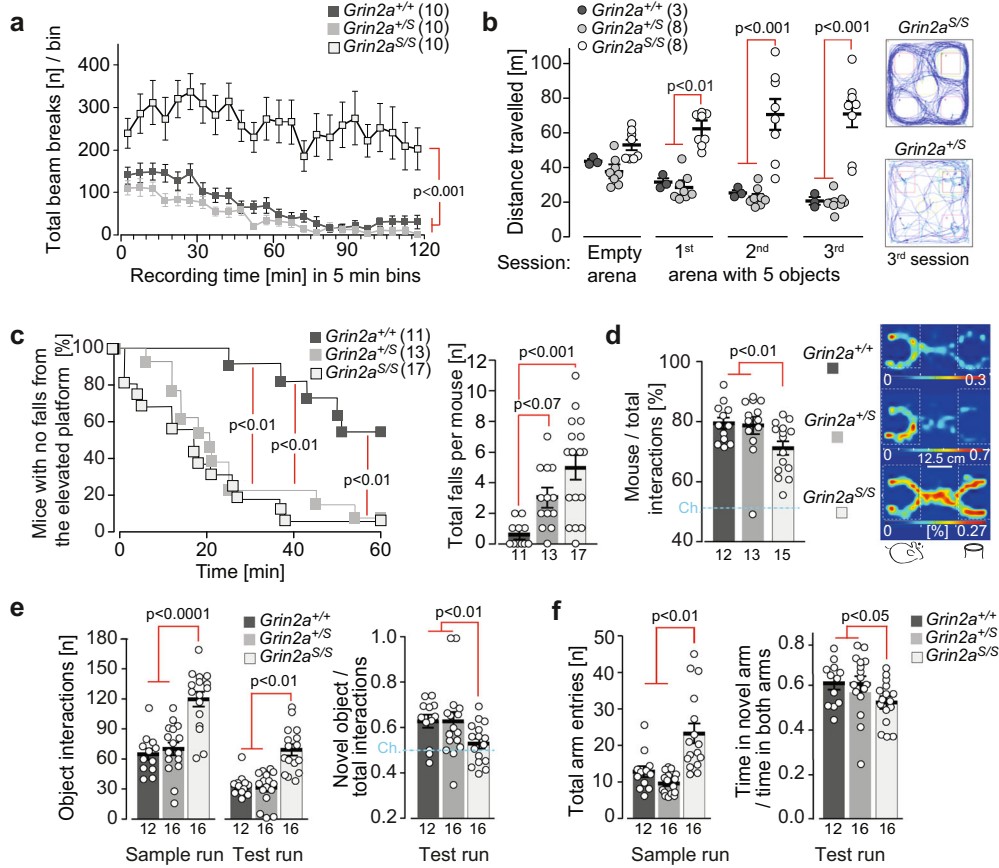

**Fig. 7 *Grin2a^{S/S}* mice are hyperactive and show increased attention and lack of inhibition. a** The total locomotor activity during the entire 2 h session in a novel environment was enhanced in *Grin2a^{S/S}* mice. **b** Similarly, the exploratory behavior of *Grin2a^{S/S}* mice remained high in all three successive 6 min exposure sessions to five novel objects (ITI = 4 min), as exemplified by traces of the animal's movement on the right. **c** (left) The time of first fall (Cliff Avoidance Reflex, CAR) was significantly reduced for *Grin2a^{S/S}* and *Grin2a^{+/S}* mice. **c** (right) In a 60-min session *Grin2a^{S/S}* mice showed a significantly increased number of falls compared to wild-type mice. **d** (left) In the three-chamber sociability test *Grin2a^{S/S}* mutants had a significantly reduced preference to explore the stimulus mouse over the object compared to *Grin2a^{+/S}* and *Grin2a^{+/+}* littermates. **d** (right) Occupancy heat maps in the three-chamber sociability test show a representative example of a *Grin2a^{S/S}* mouse with a reduced preference for another mouse versus an inanimate object. The social preference is visible for *Grin2a^{+/+}* and *Grin2a^{+/S}* mice. The occupancy is color-coded separately for each group and translates to a % given in white numbers on the key for each genotype. **e** During the novel object recognition test, *Grin2a^{S/S}* mice displayed significantly more interactions with the two objects in both the sample and test runs compared to their *Grin2a^{+/S}* and *Grin2a^{+/+}* littermates. During the test run, only the *Grin2a^{+/S}* and *Grin2a^{+/+}* showed a significant preference for the novel object over the familiar object. There was no significant novelty preference in the *Grin2a^{S/S}* mice. **f** During the sample run in the Y-maze *Grin2a^{S/S}* mice made significantly more total arm entries. In the test run, *Grin2a^{+/+}* and *Grins2a^{+/S}* showed a preference for the unexplored novel Y-maze arm, whereas *Grin2a^{S/S}* mice showed a lack of novelty preference. The numbers of mice are shown in brackets or below the bars of bar graphs. Chance levels (Ch.) are indicated by dashed lines. Error bars represent mean ± SEM (for statistics: Supplementary Statistics to Fig. 7).

higher compared to controls, whereas a similar but non-significant trend was noticed for *Grin2a^{+/S}* animals (Fig. 7c).

Next, we assessed the general sociability of the mice in the three-chamber social test[70]. The preference to interact with an unfamiliar stranger mouse rather than with a non-living novel object was significantly reduced in *Grin2a^{S/S}* mice compared to the other two genotypes (Fig. 7d). However, this reduced preference did not reflect a lack of interest in the social stimulus per se, but rather an increased activity and exploratory response to the object. Indeed, *Grin2a^{S/S}* mice showed increased levels of exploration and attention to both the social and non-social cues. Similarly, in the novel object recognition task, the total number of interactions with objects was significantly increased in *Grin2a^{S/S}* mice; however, they did not show a significant preference for the novel object over the familiar one in the test trial, as observed in wild-type controls and *Grin2a^{+/S}* littermates (*Grin2a^{+/+}* p = 0.001, *Grin2a^{+/S}* p < 0.01; *Grin2a^{S/S}* p > 0.20; Fig. 7e).

To evaluate short-term spatial recognition and spatial exploration we conducted a simple, spatial novelty preference test in a perspex Y-maze (Fig. 7f). Mice were allowed to explore freely two arms of the maze (start and familiar arm) during a sample run, whereas in the following test run they were allowed to explore all three arms (including the previously unvisited and hence novel arm). As expected, *Grin2a^{+/+}* controls showed a strong preference for the novel arm over the familiar arm during the test run, as did the heterozygous *Grin2a^{+/S}* mice. In contrast, the homozygous *Grin2a^{S/S}* mice showed reduced novelty preference (Fig. 7f) and increased locomotor activity (with significantly more total arm entries) throughout both the sample and test runs.

In summary, this behavioral analysis revealed that *Grin2a^{S/S}* mice (but not the *Grin2a^{+/S}* littermates) displayed increased and dysregulated levels of attention and exploration to spatial, non-spatial, and social cues

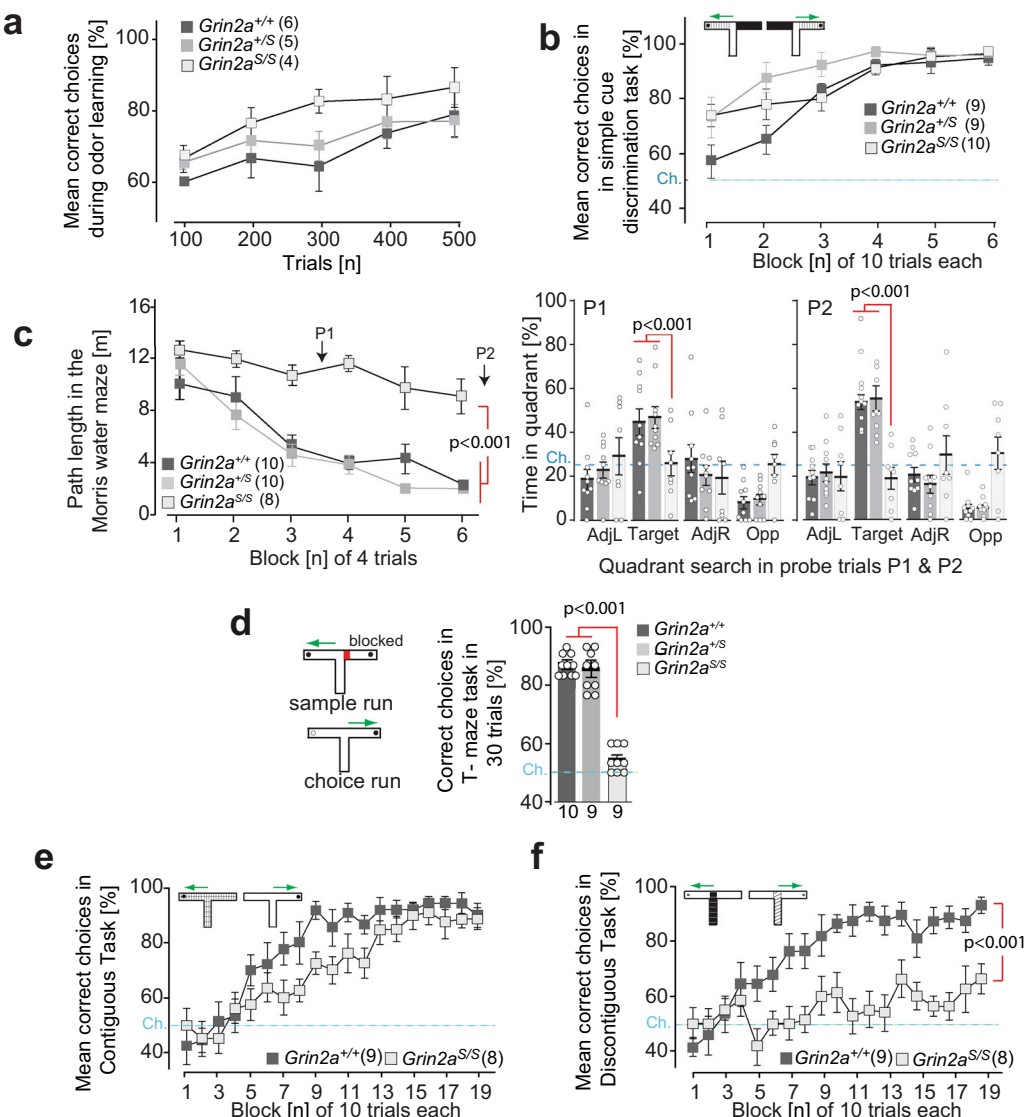

**Fig. 8 Associative learning in GluN2A(N615S) expressing mice. a** Odor discrimination. *Grin2a^{S/S}* learned to discriminate between amyl acetate and ethyl butyrate in a rewarded go/no go-paradigm (250 trials for each odor stimulus, pseudo-randomized, and counterbalanced by stimulus identity across animals). Acquisition was similar for all three genotypes. **b** Simple visuo-tactile discrimination. *Grin2a^{S/S}* mutant mice showed normal acquisition in the simple discrimination T-maze task and were able to associate a specific visuo-tactile insert (black foam versus light blue toweling) with a milk reward. **c** Morris watermaze. In the standard Morris watermaze task (left graph) the path length to reach the hidden platform decreased across training blocks for *Grin2a^{+/+}* and *Grin2a^{+/S}* mice but not for *Grin2a^{S/S}* mice. Two probe trials were conducted after 12 and 24 training trials (P1 and P2, respectively) during which the platform was removed from the pool. At both P1 and P2 (right bar graphs) the *Grin2a^{S/S}* mice failed to search for the platform in the target quadrant. AdjL adjacent left, Target fixed location of the hidden escape platform during acquisition, AdjR adjacent right, Opp opposite of target quadrant. Dashed lines indicate chance levels. **d** Rewarded alternation. (left) In the T-maze rewarded alternation task (right) spatial working memory performance was substantially impaired in *Grin2a^{S/S}* mice. **e** Contiguous task. In the contiguous version of the conditional T-maze task (with floor inserts covering the entire T-maze; white Perspex versus gray wire mesh), *Grin2a^{+/+}* and *Grin2a^{S/S}* mice were able to associate a particular floor insert with the location of the reward in either the left or the right goal arm. **f** Discontiguous task. Separate groups of mice were trained in the discontiguous version of this conditional task, in which the floor insert cues were now limited to the start arm only. *Grin2a^{+/+}* mice readily acquired the task, but *Grin2a^{S/S}* mice failed to learn. The numbers of mice are shown in brackets or below the bars of bar graphs. Chance levels (Ch.) are indicated by dashed lines. Error bars represent mean ± SEM (for statistics: Supplementary Statistics to Fig. 8).

**Grin2a^{S/S} mice are not impaired in learning simple associations.** How does this attentional phenotype impact associative learning? We found that despite their hyperactivity, impulsivity and dysregulated levels of attention, *Grin2a^{S/S}* mice could still form simple associations. *Grin2a^{S/S}* and *Grin2a^{+/S}* mice were both able to learn to discriminate between a rewarded and a non-rewarded odor, and both groups of mutants acquired this simple odor-learning task at the same rate as their control littermates (Fig. 8a).

We next evaluated the ability of mice to acquire a simple visuo-tactile discrimination in a T-shaped maze, during which animals were required to choose between two different floor inserts located in the goal arms (Fig. 8b). Choosing the correct floor insert (whose spatial location varied between the left and right goal arms according to a pseudorandom sequence such that there was no spatial solution) was rewarded with a sweet milk reward. Both *Grin2a^{S/S}* and *Grin2a^{+/S}* mice successfully acquired the task and were unimpaired relative to controls (Fig. 8b). *Grin2a^{+/S}*

mice were slightly faster at acquiring the task than controls ($p <$ 0.05). Once the animals had acquired this simple visuo-tactile discrimination, we then reversed the floor-insert cue–reward contingencies for each mouse (thus animals that had been trained on A+/B− now experienced A−/B+ and vice-versa). There was no effect of genotype during reversal learning, with all three groups of mice acquiring the new cue–reward contingencies at the same rate (Supplementary Fig. 7a). Thus, $Grin2a^{S/S}$ mice could form simple cue/reward associations using cues from different sensory modalities, and could also perform more complex, cognitively demanding tasks using these cues, for example, when cue–reward contingencies were reversed.

**$Grin2a^{S/S}$ mice are impaired on a battery of spatial memory tasks**. We then assessed the spatial learning abilities of mice across a battery of spatial memory tasks. First, we investigated spatial reference memory (SRM) acquisition in the Morris watermaze (MWM)[71], during which mice were trained to find a hidden platform that remained in the same, fixed location on every trial, starting from different points around the perimeter of the pool. Using the same protocol, with the same extramaze spatial cues and in the same testing laboratory, we have previously shown that the acquisition of this task is dependent on the hippocampus[72,73]. Whereas both $Grin2a^{+/+}$ littermates and $Grin2a^{+/S}$ mice readily learned the location of the platform with training, $Grin2a^{S/S}$ mice displayed a large impairment, showing only a marginal improvement across training sessions (Fig. 8c and Supplementary Fig. 7b). In the probe tests during which the platform was removed from the pool (performed after 12 and 24 training trials), the $Grin2a^{+/S}$ and $Grin2a^{+/+}$ littermates showed a strong and robust preference for the training quadrant that normally contained the escape platform. In contrast, the $Grin2a^{S/S}$ mice had no memory for the platform location and exhibited chance levels of performance in both probe test trials (Fig. 8c), in marked contrast to the successful acquisition that we have previously reported with $Grin2a^{-/-}$ mice in this very same MWM task[74].

To exclude the possibility that the impaired MWM performance of $Grin2a^{S/S}$ mice was caused by deficits in motivation or sensorimotor abilities, a visible platform version of the task was also conducted. In this task, the platform was raised above the water level and clearly marked with a black and white striped cylinder directly above. Mice of all three genotypes rapidly learned to swim towards the visible platform and all mice reached nearly identical levels of performance by the third block of testing (Supplementary Fig. 7c), consistent with our results from the simple associative tasks using either odor- or visuo-tactile cues (Fig. 8a, b). Thus, the spatial learning impairment in the MWM was unlikely due to sensorimotor or motivational disturbances.

To test the generality of this spatial learning deficit in a completely different experimental setting, with different sensorimotor and motivational demands, we next assessed acquisition of an appetitively motivated, SRM six-arm radial maze task. Here, $Grin2a^{S/S}$ mice showed a substantial deficit in learning to discriminate between the rewarded and non-rewarded arms. While both the $Grin2a^{+/S}$ and $Grin2a^{+/+}$ littermates learned the locations of the food rewards and showed a gradual improvement across days of training, making less reference memory errors as training progressed, the $Grin2a^{S/S}$ mice showed little, if any, improvement (Supplementary Fig. 7d). Spatial working memory (SWM) was also assessed using both the spontaneous and rewarded alternation versions of the T-maze task[75,76]. In both tasks, the $Grin2a^{S/S}$ mice were substantially impaired relative to $Grin2a^{+/S}$ and $Grin2a^{+/+}$ littermates (spontaneous: Supplementary Fig. 7e; rewarded: Fig. 8d).

**$Grin2a^{S/S}$ mice are impaired when there is spatiotemporal discontiguity**. A potential feature of these spatial memory tasks which differentiates them from the simple associative learning tasks is the presence of a spatiotemporal discontiguity between the relevant environmental cues at the point at which a decision is made (e.g. to navigate in a particular direction) and the occurrence of the unconditioned stimulus (e.g. the escape platform in the watermaze, food rewards on the radial maze). To determine specifically whether $Grin2a^{S/S}$ mice were sensitive to spatiotemporal discontiguities we returned to the appetitively motivated T-maze task, in which the presence of visuo-tactile cues in the form of floor inserts in the maze determined the location of a sweet milk reward. We have shown that $Grin2a^{S/S}$ mice could successfully acquire a simple visuo-tactile discrimination during which floor inserts restricted to the goal arms of the maze indicated the presence or absence of reward (Fig. 8b). We next used the entire floor structure of the T-maze as a conditional cue to indicate whether the reward was located in either the left or the right goal arm (Fig. 8e). Mice had to follow the rule that if floor insert A is present then reward is on the left whereas if floor insert B is present then reward is on the right. When the floor inserts covered the entire T-maze, including the start arm and both goal arms, such that there was no discontiguity, then both $Grin2a^{+/+}$ and $Grin2a^{S/S}$ mice were able to acquire the task to a high level of performance by the end of 19 blocks of training (Fig. 8e). There was a subtle impairment in the $Grin2a^{S/S}$ mice such that they acquired the task slightly more slowly than the controls, but by 130 trials both groups were consistently scoring over 80% correct (Fig. 8e). Separate groups of $Grin2a^{+/+}$ and $Grin2a^{S/S}$ mice were then trained on the discontiguous version of the task in which the floor insert cues were now limited to the start arm only and thus absent from the goal arms and at the point when the mouse received the reward (Fig. 8f). This time, while wild-type littermates readily acquired the task, $Grin2a^{S/S}$ mice failed to learn, showing a significant impairment from training blocks 5 to 19. Thus, the $Grin2s^{S/S}$ mice were able to acquire the T-maze task but only when there was no spatiotemporal discontiguity present. Importantly, there was a highly significant genotype by task interaction ($F(1, 30) = 5.1$; $p < 0.05$), and subsequent analysis of simple main effects showed that there was a significant effect of discontiguity for the $Grin2a^{S/S}$ mice ($F(1, 30) = 13.2$; $p < 0.005$), but not for the wild-type controls ($F < 1$; $p > 0.20$).

In summary, our behavioral analysis showed that $Grin2a^{S/S}$ mice could form simple associations and thus perform discrimination tasks using cues from different sensory modalities, including during a more cognitively demanding reversal learning task. However, the same $Grin2a^{S/S}$ animals were considerably impaired in a battery of spatial learning paradigms, although this was not the case for heterozygous $Grin2a^{+/S}$ littermates. $Grin2a^{S/S}$ mice were also impaired on conditional T-maze tasks, particularly when there was a spatiotemporal discontiguity between the relevant sensory cues present at the choice point and the presence/absence of reward.

## Discussion

Here we provide the first direct experimental evidence that the tightly controlled $Mg^{2+}$ block of the GluN2A-type NMDAR, and thus the voltage-dependent $Ca^{2+}$ influx through its ion channel, is essential in different regions of the nervous system for coordinated neuronal communication and its functional consequences.

Our data obtained both on recombinant NMDARs and on NMDARs from genetically modified animals shows that GluN2A (N615S) containing NMDARs have a remarkably reduced sensitivity to external $Mg^{2+}$ ions as measured by the linearized $I/V$ relation of GluN1/2A(N615S) channels and their less reduced

voltage-controlled NMDARs currents in the presence of $Mg^{2+}$ in HEK293 cells and hippocampal slices, respectively. Taking into account the 1.4-fold increased $Ca^{2+}$ permeability[15], no alterations in the channel conductance and proven surface and synaptic expression of GluN2A(N615S)-containing NMDARs, all synaptic events are most likely associated with an increased NMDAR-mediated $Ca^{2+}$ entry at resting membrane potentials through synaptic channels. Moreover, spill-over of glutamate might produce additional $Ca^{2+}$-influx through GluN1/2A(N615S) receptors into dendrites located close to active synapses and will amplify the impact of volume transmission to the network activity and the disturbance of synaptic plasticity in GluN2A(N615S)-expressing mice.

Alteration in synaptic plasticity could be monitored by an increased LTP component of GluN2B-containing NMDARs in hippocampi from GluN2A(N615S)-expressing mice. Very similar to our findings of an increased GluN2B component at CA3-to-CA1 synapses in young mice[34], the LTP could be significantly enhanced by repeated tetanic stimulations and the enhancement could be blocked by CP101,106 in adult $Grin2a^{+/S}$ and $Grin2a^{S/S}$ mice. This might indicate that the altered $Ca^{2+}$ homeostasis by the glutamate-triggered $Ca^{2+}$ influx through the mutant NMDARs keeps the plasticity mechanism of CA1 cells in an immature state in GluN2A(N615S)-expressing mice.

The homozygous $Grin2a^{S/S}$ mice exhibited a complex phenotype characterized by high sensitivity to AGSs and cognitive deficits. In response to a strong acoustic stimulus, unregulated GluN2A receptor-mediated $Ca^{2+}$ signaling is associated with over-excitation in the midbrain/brainstem, followed by strong activation of some forebrain neuronal populations (mainly in the amygdala and hypothalamus), a response that cannot be compensated for and finds its end in generalized epileptic seizures with respiratory arrest (AGS-RA). In contrast, the hippocampal formation appears less excitable in these mice and hippocampal theta frequency and phase coupling between slow oscillations and the amplitude of a fast oscillation is reduced during exploration. These oscillatory changes correlated with impaired regulation of exploratory activity and attention, resulting in associative learning deficits in situations in which the tight regulation of attentional processes is likely to be important. Thus, our analysis of GluN2A(N615S)-expressing mice provides strong and direct evidence for the divergent and brain region-specific role of GluN2A-type NMDAR $Ca^{2+}$ signaling in the excitability of neuronal networks. Opposite to what was observed previously for $Grin2a^{-/-}$ mice[74], the voltage-independent GluN2A-type NMDAR $Ca^{2+}$ signaling in $Grin2a^{S/S}$ mice generated increased excitability in neurons of midbrain nuclei and decreased excitability of hippocampal principal neurons.

The AGS activity pattern in $Grin2a^{S/S}$ mice shows strong similarity to the one described in subcortical/midbrain structures of epilepsy-prone DBA mice and GEPR-9s rats, which are frequently used as animal models for sudden unexpected death in epilepsy[77]. In those animal models, which rely on acoustic kindling, the IC plays a key role in the initiation of epileptic seizures, which also involves non-auditory brain structures such as the PAG and the substantia nigra pars lateralis[78], consistent with our c-Fos expression analyses of $Grin2a^{S/S}$ brains after AGS. The central role of the IC is supported experimentally by the increased AGS sensitivity seen in normal rats after inhibition of $GABA_A$ receptors in IC or after pharmacological NMDAR activation of this nucleus[79]. EEG recordings and c-Fos expression mapping have demonstrated that kindling causes permanent AGS network expansion from the brainstem to forebrain structures, most likely via the amygdala[47,78,80], increasing the severity of the seizures, and supported also by our results here in $Grin2a^{S/S}$ mice. In line with previous experiments[81], we were able to effectively block AGS in $Grin2a^{S/S}$ mice by pre-treatment with NMDAR antagonists, providing evidence that the NMDAR signaling directly promotes the brainstem-derived AGS induction. Notably, $Grin2a^{-/-}$ mice are more resistant to electrical IC stimulation,[50] revealing a key role for GluN2A-type receptors in the modulation of IC neuronal activity. Furthermore, these data indicate that in the absence of GluN2A-type NMDARs, IC neurons are either less excitable and/or under stronger inhibition. In $Grin2a^{S/S}$ mice, we have found exactly the opposite situation: the less-stringent voltage-controlled $Ca^{2+}$ influx through GluN2A-containing NMDAR leads to an over-excitation and/or reduced inhibition of the neuronal populations that are activated by the audiogenic stimulus and trigger AGS. Thus, the constant glutamate-gated GluN1/2A(N615S) $Ca^{2+}$ influx seems to mimic experimental "kindling" models in sensitizing the network response to a strong acoustic stimulus. MK-801 and memantine have an AGS protective effect in kindling models as well as in $Grin2a^{S/S}$ mice. Thus, the AGS network response is most likely triggered predominantly by voltage-controlled NMDAR signaling through GluN1/2B or other GluN2A-independent NMDAR subtypes. A drug effect on GluN1/2A(N615S) receptors seems less likely, since MK-801 binding and memantine blockade of recombinant GluN1/2A(N615K) receptors is reduced[55].

Opposite and contrasting results between $Grin2a^{-/-}$ and $Grin2a^{S/S}$ mice were also found when analyzing MK-801 c-Fos activity mapping in the hippocampus. In contrast to the IC neurons, the glutamate-gated GluN1/2A(N615S) $Ca^{2+}$ influx was associated with reduced excitability of hippocampal principal cells. Although in $Grin2a^{S/S}$ mouse brains MK-801-induced c-Fos expressing cells were increased in several cortical areas, they were much less visible in the hippocampus, in line with the lack of hippocampal activation observed after AGS. This suggests that the hippocampus of $Grin2a^{S/S}$ mice is less active, which might be due to increased activity of GABAergic interneurons[56], or could reflect wider circuit effects. In contrast, in the absence of GluN2A in $Grin2a^{-/-}$ mice, the hippocampal pyramidal and granular neurons are highly excitable following MK-801 challenge. Thus, the regulation of NMDAR $Ca^{2+}$ signaling via the voltage-dependent $Mg^{2+}$ block plays a key role in the differential priming of neural activity in separate and distinct neural networks.

Changes in excitatory activity in the hippocampus of behaving $Grin2a^{S/S}$ mice were associated with impaired neuronal network activity. We noticed a reduced phase–amplitude cross-frequency coupling in the lower gamma frequency range during exploration and REM sleep, and a reduced theta frequency peak during exploration, which could potentially be explained by the altered inhibition/excitation ratio of the neuronal networks[82]. This is in line with previous studies in which NMDAR ablation in hippocampal parvalbumin-positive (PV+) interneurons showed disturbed theta-nested gamma oscillations[83–87]. Interestingly, whereas NMDAR blockade leads to hypersynchronous phase locking[88], we observed the opposite when introducing the $Grin2a^{(N615S)}$ mutation. Thus, disrupted coincidence detection of the GluN2A-type NMDARs affects the coordinated oscillatory network activity and the slow inhibition of the hippocampal network during theta-related behaviors.

The hippocampal theta rhythm is of particular importance during exploration and attention[89,90]. Notably, we observed clear behavioral impairments during exploration, including hyperactivity and unregulated and exaggerated levels of attention to spatial, non-spatial and social cues in the $Grin2a^{S/S}$ mice. Despite this, $Grin2a^{S/S}$ mice could still form simple associations in several different tests, but only if the relevant sensory cues and the rewards were contiguous in space and time. Thus, basic

associative learning mechanisms remained intact in Grin2a$^{S/S}$ mice, in accordance with the behavior of Drosophila over-expressing transgenic Mg$^{2+}$ insensitive NMDARs[13].

Analogous to the above-mentioned transgenic flies, Grin2a$^{S/S}$ mice did exhibit pronounced learning deficits during a battery of spatial memory tests (e.g. MWM and radial maze). They were also impaired on the conditional T-maze learning task[91], particularly when the relevant sensory cues at the choice point were separated in space and time from the reward outcomes. This might reflect that the Grin2a$^{S/S}$ mice exhibit elevated, sustained, and unregulated levels of attention, which can be particularly disruptive for tasks with inherent spatiotemporal discontiguity during which appropriate and specific cues must be held in the forefront of attention to span or bridge any gaps between relevant cues and their associated outcomes. Paying attention to irrelevant or inappropriate cues might be very disruptive on such tasks. This is in contrast to simple associative learning tasks, during which sensory cues and rewards/outcomes are present at the same time and in the same place, and thus credit assignment and appropriate associations can be formed unimpaired. Thus, the strictly controlled voltage-dependent Ca$^{2+}$ signaling of GluN2A-type NMDARs appears to be of particular importance for the tight temporal control of attentional processes which becomes especially important when there are spatial and/or temporal discontiguities between relevant cues and behaviorally relevant outcomes.

In recent years numerous de novo GRIN2 variants have been identified in patients with neurological disorders. One set of rare mutations changes the activation profile of NMDARs and is located in the ligand-binding domain (LBD) of the NMDAR. A second set is located in the NMDAR channel pore of the channel gate[92]. These mutations affect ion permeability and in particular the voltage-controlled Ca$^{2+}$ influx. Carriers of these second set of mutations suffer from neurological dysfunction with different degrees of severity. Up to 500 rare disease variants in NMDAR genes have been identified in human patients. Most variants were found in GRIN2A and GRIN2B. The comparison of 249 individuals with pathogenic, or likely pathogenic, GRIN2A variants identified patients with severe developmental phenotypes associated with missense mutations in the ion pore or the linker domain. NMDAR mutations within the amino-terminal, LBD, and null variants led to a less severe phenotype and were classified as "loss of function mutations"[93,94] whereas most of the severe NMDAR mutations with altered Mg$^{2+}$ block were considered as "gain of function" mutations[94].

A detailed study with 12 de novo GRIN missense variants from 18 patients clearly showed that those missense mutations in the P2 loop of GluN1, GluN2A, and GluN2B altered surface expression, pharmacological properties, and other biophysical characteristics. It also demonstrated that these variants can have modest changes in agonist potency and proton inhibition. Furthermore, the voltage-dependent inhibition by Mg$^{2+}$ was significantly reduced in all variants. Since the single channel conductance and Ca$^{2+}$ permeability can be altered to different extents by different mutations, the degree of Ca$^{2+}$ influx after glutamate stimulation is specific for each mutation. This means that the severity of any "gain of function" mutation (like the Grin2a$^{(N615S)}$) and thus the severity of the associated phenotype will be defined by the kind of mutation[95]. Li et al. summarize the phenotypes of 4 GluN2A(N614S), 2 (N615K), and 1 GluN2B (N615I), 1 (N615K) and 1(N616K) de novo mutations. Seven of these nine patients showed muscle hypertonia, six suffered from epileptic seizures, and all exhibited ID and developmental delay. All GluN2A mutations were associated with language problems and autism spectrum disorder was diagnosed in one GluN2A (N614S) patient and in one GluN2B(N615I) carrier. This is in line

with other studies in heterozygous patients with the very same de novo NMDAR mutations who displayed a similar (although not identical) clinical manifestation[92,94–96].

Indeed, although two patients with the very same GluN2A (N615K) mutation had early-onset epileptic encephalopathies, there were also differences in their clinical manifestations which defied a clear genotype–phenotype correlation[26]. Correspondingly, in our heterozygous Grin2a$^{+/S}$ mice the gain of function Grin2a$^{(N615S)}$ mutation led—depending on the behavioral test and sometimes on the individual mouse—to either detectable or non-detectable phenotypes (for phenotypes that were clearly evident in the homozygous mutant mice). Since glutamate stimulation—and thus neuronal activity—is an obligatory trigger for Ca$^{2+}$ influx through mutated NMDARs, these gain of function NMDAR mutations may be particularly sensitive to non-genetic-factors that affect neuronal activity.

To conclude, our experimental findings thus provide a detailed physiological explanation for the cognitive impairments combined with epileptic seizures, hypotonic muscle tone, and developmental delay in patients carrying similar de novo NMDAR mutations GRIN2A$^{(N615K)}$, GRIN2B$^{(N615I)}$, and GRIN2B$^{(V618G)}$. Notably, all three of these GluN2 subunit mutations were found to reduce the Mg$^{2+}$ block and to alter Ca$^{2+}$ permeability in recombinant systems[27,97]. Our results demonstrate that the tightly regulated Mg$^{2+}$ block of the NMDAR, and thus the voltage-controlled Ca$^{2+}$ influx, is essential in different regions of the nervous system for coordinated neuronal communication. Consequently, its disruption has profound consequences on brain function in numerous different domains.

## Methods

**Ethical statement.** Most animal experiments were performed at the Max Planck Institute of Medical Research in Heidelberg according to the institutional guidelines of the Max Planck Society, the "Interfakultäre Biomedizinische Forschungseinrichtung" (IBF) animal core facility of Heidelberg University and the Interdisciplinary Neurobehavioral Core (INBC) of Heidelberg University. Genetic manipulations of mice were performed under the licenses of the Regional Board Karlsruhe, Germany: Generation of mice (35-9185.81/G-4/02); mouse behavior and drug treatment of animals (35-9185.81/G-115/04, 35-9185.81/G-71/10; 35-9185.81/G-171/10; 35-9185.81G-105/16); in vivo (35-9185.81/G-273/12; 35-9185.81/G-171/10, 35-9185.81/G-44/16, 35-9185.81/G-100/16). Animals for molecular, histological, and electrophysiological experiments were recorded under the protocols MPI/T-6/06; 15/08; 20/9, 28/11. Ex vivo LTP experiments were conducted according to the Norwegian Animal Welfare Act and the European Union's Directive 86/609/EEC. Behavioral experiments in the UK were conducted in accordance with the United Kingdom Animals Scientific Procedures Act (1986), under the project license number PPL 30/2561 of the UK Home Office.

**Statistics and reproducibility.** In all figures, the number of independently recorded values is clearly indicated. The number of animals and the number of recorded data points are given as appropriate. In bar graphs, all data points used are pictured together with the standard error of the mean. In the figure legends, the tests used for statistical evaluation (ANOVA, t-test, etc.) are stated together with the p values of the results. P values indicating a significant difference are given directly in the figures. Due to space limitations in the main figure legends, detailed descriptions of the statistical analyses can be found in Supplementary Statistics to Figs. 1, 2, 3, 4, 7 and 8. In the Supplementary Figures, the details of the statistical analyses are given in the figure legends. When multiple comparisons were used to control the familywise error rate, we indicate the statistical test used (e.g. Bonferroni test). When appropriate, non-parametric analyses (e.g. Mann–Whitney U-test) were conducted. Data records in vivo local field potential oscillations can be provided on request. Data were first recorded and analyzed in Excel (Microsoft), and for the distribution blots, Graphpad Prism was used.

**Generation of GluN2A(N615S)-expressing mice.** For gene targeting at the Grin2a gene loci, we used the method which we described in detail for the targeting of the Grin1 gene[22,28]. A brief description of the targeting strategy to generate Grin2a$^{tm1.RSP}$ $^{(N615S)}$ mice (herein abbreviated as Grin2a$^{S/S}$) is given in Supplementary Fig. 1. Mice are genotyped by tail-PCR with specific primers. Primers used were: do: 2A-TM3do (5′-GTG TGG GCC TTC TTT GCY GTC-3′) and up: 2A-IN11UP1 (5′-CAT ATA TAC AAG CAT TGG AG-3′). Amplified gene fragments for the Grin2a$^{S}$ and Grin2a$^{+}$ alleles are 559 and 482 bp, respectively. The Grin2a$^{S/S}$ mice were used

previously in a cellular LTP experiment induced by a low-frequency stimulation pairing protocol[30]. The mouse line is available from the INFRAFRONTIER'S EMMA mouse respiratory (EM:09319: *B6.129-Grin2a$^{tmN615SRSP}$/kctt*).

**Expression analysis of Grin2a mRNA, cDNA synthesis, reverse transcription PCR, and sequencing**. Mice were killed by decapitation, the total brain was immediately isolated, and the forebrains (excluding cerebellum and olfactory bulb) were used as tissue input for the cDNA preparation using the RNeasy mini Kit (Qiagen, CD./ID: 74104). From the forebrain lysate (about 20 mg), total RNA was prepared using the RNeasy spin column, and the remaining DNA was digested by DNAseI. First-strand cDNA synthesis was performed from about 1.5 µg total RNA using a Superscript II reverse transcriptase kit (Superscript$^{TM}$ III First-Strand Synthesis System for RT-PCR. Cat. No 18080-51; now Thermo Fisher) primed by random hexamers according to the manufacturer's instructions. From this reaction, 2 µl was directly used to amplify the M2 region spanning *Grin2a* cDNA fragment by the Grin2a mRNA-specific primers NR2Aex9do (5′-CCT GTT GGA TAC AAC AGA AAC TTA GC-3′) and NR2Aex11up (5′-CTG GTT GAA TTT GGT CAT GTA CTG-3′) in 30 PCR cycles (Phusion$^{®}$ High-Fidelity PCR Kit; New England Biolabs NEB; E0553S/L). The amplified 390 bp DNA fragment was gel-purified on a 1.5% agarose gel in E-buffer (QIAquick Gel Extraction Kit; Qiagen, CatNo./ID: 28704) and primer NR2Aex11up was used for sequencing using a commercial service (Eurofins, GATC Service). Sequence analyses were performed by Lasergene (DNASTAR) and the chromatograms of the reverse strands (coding sequence) were visualized with SnapGene viewer (SnapGene).

**Electrophysiological profile of GluN1/2A(N615S) receptors in HEK293 cells**. For the transfection of human embryonic kidney cells (HEK293 cells; ATCC: CRL-1573$^{TM}$) we used the $Ca^{2+}$ phosphate precipitation method of Chen and Okayawa[98]. HEK293 cells were co-transfected with plasmids expressing the rat GluN1-1a[99] together with GFP and the rat GluN2A[100] or GluN2A(N615S)[14]. Two days after transfection, GFP-labeled cells were lifted from the coverslip and whole-cell currents were recorded using an EPC-9 amplifier (HEKA, Lambrecht, Germany) in the presence of 10 µM glycine by fast applying 1 mM glutamate from a Piezo-driven double-barrelled pipette. Solution exchange time, measured with an open patch pipette, was 100–200 µs. The extracellular solution contained (in mM): 135 NaCl, 5.4 KCl, 1.8 $CaCl_2$, 5 HEPES (pH 7.3). Whole-cell currents were activated at holding potentials between −100 and +40 mV, in steps of 20 mV, in the presence of 0, 1, or 4 mM $MgCl_2$[101]. Activation/deactivation and desensitization kinetics were determined at −60 mV applying glutamate for 20 and 600 ms, respectively. Patch pipettes (3–5 MΩ) were filled with (in mM): 140 CsCl, 2 MgATP, 10 EGTA, 10 HEPES (pH 7.3, 290–305 mOsm). Statistical significance between wild type and mutated GluN2A was evaluated by an unpaired Student's *t*-test.

**Immunoblot analysis**. Forebrains were placed in a glass pestle into ice and 1 ml of cold homogenization buffer (0.32 M sucrose, 25 mM HEPES pH 7.4, 0.5 mM EDTA, protease and phosphatase inhibitors) was added to homogenize the tissue using 10–15 strokes, avoiding the formation of bubbles. Then, the homogenate was centrifuged at 2000 r.p.m. at 4 °C to remove the pellet nuclear fraction (P1). The supernatant (S1 = total lysate) was centrifuged at 14,000 r.p.m. for 30 min at 4 °C to yield crude cytosol (S2) and a crude membrane pellet (P2). Pellet 2 was resuspended in lysis buffer (1 M Tris-HCl pH 7.6, 5 M NaCl, 1 M KCl, 10% Triton X-100, 10% Nonidet P40, 0.5 M EDTA, protease, and phosphatase inhibitors). The protein concentration of the samples was measured by Bradford protein assay. Protein samples (10 µg/lane) were separated by SDS/PAGE and analyzed by standard immunoblotting[102]. Antibodies used anti-GluA1 (1:2000, AB1504; Millipore; RRID: AB_2113602), anti-GluN1 (1:1000; AB_9864R; Millipore RRID: AB_10807557); anti-GluN2A (1:2000; M264; Sigma-Aldrich, RRID: AB_260485); anti-GluN2B (1:1000; MAB5574; Millipore, RRID:AB_11213424); anti-PSD95 (1:2000; AB9708; Millipore, RRID:AB_11212529); anti-P-CaMKII (1:2000; MA1-047; Thermo Scientific, RRID:AB_325402); anti-GAPDH (1:5000, AB8245; Abcam, RRID:AB_2107448); anti-beta actin (1:15,000; A5441; Sigma-Aldrich, RRID: AB_476744); secondary goat anti-rabbit (RRID:AB_2336198) and goat anti-mouse (1:15,000, Vector; RRID:AB_2336171). Immunoreactivity was detected with ECLplus (GE Healthcare) and immunoblots were scanned and quantitatively analyzed with ImageJ. Western blot quantification was statistically evaluated by one-way ANOVA followed by Tukey's multiple comparisons tests. Data are presented as mean ± SEM.

**Single-cell electrophysiology**. For synaptic recordings in acute hippocampal slices, the brain was dissected out in ice-cold artificial cerebrospinal fluid (aCSF) (in mM: 125 NaCl, 2.5 KCl, 25 glucose, 25 NaHCO₃, 1.25 $NaH_2PO_4$, 2 $CaCl_2$, and 1 $MgCl_2$), which was saturated with 95% $O_2$/5% $CO_2$. Neocortical slices (transverse; 250-µm-thick) were cut on a vibratome. A hemisphere was glued at the surface of the sagittal plane onto a block, which was mounted at a 10° angle such that the blade cuts from the upper border of the neocortex toward the caudal border and down towards the midline. Slices were incubated for 30 min at 34 °C and then at room temperature (22–25 °C) until they were transferred to the recording chamber (22–25 °C). In the acute hippocampal slices, pyramidal CA1 neurons were

identified visually using IR-DIC microscopy. Identified cells were clamped at −70 mV[35,103], either in the presence or in absence of 1 mM $MgCl_2$. To evoke synaptic currents, glass electrodes filled with aCSF were placed in the stratum radiatum within 50–100 µm of the soma of the recorded neuron. Inhibitory synaptic transmission was blocked during recordings by the addition of 10 µM gabazine to the perfusion aCSF. The inter-sweep interval was 6 s. AMPAR- and NMDAR-mediated currents were pharmacologically dissected using the AMPAR and NMDAR antagonists, 2,3-dihydroxy-6-nitro-7-sulfamoyl-benzo[f]quinoxaline (NBQX; 10 µM), and (2R)-amino-5-phosphonopentanoate (AP5; 100 µM), respectively. After recording the total current responses (containing both AMPAR and NMDAR components, 100 sweeps), AMPAR channels were blocked by bath application of NBQX (10 µM) and another 100 sweeps containing only NMDAR responses were recorded. AMPA currents were obtained by subtraction of the averaged NMDA response from the averaged total response. AMPA/NMDA ratios were calculated as the peak AMPAR-mediated current amplitudes divided by the peak NMDAR-mediated current amplitudes.

**Hippocampal field LTP recordings in acute brain slices**. For local field potential recordings at hippocampal CA3-to-CA1 synapses we used transverse, acute brain slices[35,103–106]. In brief:

*Slice preparation*. Adult mice (2–4 months, 3–5 mice, per genotype and experiment) were sacrificed with Suprane (Baxter) and the brains were gently removed from the skull. Transverse slices (400 µm) were cut from the middle and dorsal portion of each hippocampus (Supplementary Data: Video 3) with a vibroslicer (Campden Instruments NVSLM1) in cold aCSF (4 °C, bubbled with 95% $O_2$–5% $CO_2$) containing (in mM): 124 NaCl, 2 KCl, 1.25 $KH_2PO_4$, 2 $MgSO_4$, 1 $CaCl_2$, 26 NaHCO₃, and 12 glucose. Slices were placed in an interface chamber exposed to humidified gas at 28–32 °C and perfused with aCSF (pH 7.3) containing 2 mM $CaCl_2$ for at least 1 h prior to the experiments. In some of the experiments, DL-2-amino-5-phosphopentanoic acid (AP5, 50 µM; Sigma-Aldrich) was added to the aCSF in order to block NMDAR-mediated synaptic plasticity or a 10 µM concentration of GluN2B-specific antagonist CP101,606 (CP) (Pfizer) was added to the perfusion media.

*Synaptic excitability*. Orthodromic synaptic stimuli (<300 µA, 0.1 Hz) were delivered through tungsten electrodes placed in either stratum radiatum proximal or stratum oriens distal of the hippocampal CA1 region. The presynaptic volley and the fEPSP were recorded by a glass electrode (filled with aCSF) placed in the corresponding synaptic layer while another electrode placed in the pyramidal cell body layer (stratum pyramidale) monitored the population spike. Following a period of at least 10–15 min with stable responses, we stimulated the afferent fibers with increasing strength. A similar approach was used to elicit paired-pulse responses (50 ms interstimulus interval, the two stimuli being equal in strength). To assess synaptic transmission, we measured the amplitudes of the presynaptic volley and the fEPSP at different stimulation strengths. In order to pool data from the paired-pulse experiments, we selected responses to a stimulation strength just below the threshold for eliciting a population spike on the second fEPSP.

*LTP of synaptic transmission*. Orthodromic synaptic stimuli (50 µs, <300 µA) were delivered alternately through two tungsten electrodes, one located in the stratum radiatum and another one in the *stratum oriens* of the hippocampal CA1 region. Extracellular synaptic potentials were monitored by two glass electrodes (filled with aCSF) placed in the corresponding synaptic layers. After obtaining stable synaptic responses in both pathways (0.1 Hz stimulation) for at least 10–15 min, one of the pathways was tetanized (either with a single 100 Hz tetanization of 1 s or repeated four times at 5 min intervals). To standardize the procedure, the stimulation strength used for tetanization was just above the threshold for generating a population spike in response to a single test shock. Synaptic efficacy was assessed by measuring the slope of the fEPSP in the middle third of its rising phase. Six consecutive responses (1 min) were averaged and normalized to the mean value recorded 1–4 min prior to tetanization. Data of the same experimental group were pooled across animals and are presented as mean ± SEM (see also ref. [35]). Statistical significance was evaluated by using a linear mixed model analysis (SAS 9.1) with $p < 0.05$ being designated as statistically significant.

*Hippocampal field LTP recordings in freely moving mice:* Local field potential at hippocampal CA3-to-CA1 synapses in freely moving mice were recorded from implanted electrodes in freely moving mice[107]. Briefly, 2-month-old mice were deeply anesthetized with a mixture of ketamine (65 mg/kg) and xylazine (14 mg/kg). In a stereotaxic frame, two mini-screws were fixed above the cerebellum serving as reference and ground electrodes respectively. Then, a bipolar stimulation electrode (two insulated tungsten wires glued together, each 52 µm in diameter, California Fine Wire) and a recording electrode (single wire, same material with stimulation electrode) were positioned in stratum radiatum by a motorized manipulator (Luigs & Neumann)[107]. Electrodes were permanently fixed with dental acrylic and surgical wounds were sutured. Singly-housed mice were allowed to recover with access to food and water ad libitum for at least 1 week before recording.

Mice were put in the recording chamber (50 cm diameter round arena, 50 cm high) for environmental acclimation overnight. A miniature headstage (1 g, npi electronic GmbH, Tamm) was connected to electrodes/pins in the presence of 95% $O_2$ containing 4% isoflurane for stress relief. Evoked LFPs were filtered with a band width of 0.3–500 Hz, amplified with the miniature headstage (EXT-02F, npi

electronic GmbH), and stored at 10 kHz (ITC-16, HEKA Elektronik) after 50 Hz noise filtration by a Hum Bug Noise Eliminator (AutoMate Scientific, Inc.). Extracellular stimulation was generated with an isolated stimulator (A365, WPI). The slopes of evoked LFPs were analyzed based on the middle one-third of the rising phase (Fitmaster, HEKA Elektronik). At the beginning of each recording, two IO curves per mouse were generated, applying stimulation voltages with both polarities. To evaluate changes in synaptic efficacy, a stimulus strength eliciting 35–40% of the maximum slope was used as the test pulse and was given every 30 s. For LTP induction, two trains of high-frequency stimulation ($50 \times 100$ Hz, 100 μs pulse width, same intensity as test pulse) separated by 5 min were used.

After recordings, mice were deeply anesthetized and electrical lesions were induced twice (20 μA, 10 s) for each single tungsten wire. Subsequently, mice were perfused with phosphate-buffered saline (PBS) followed by 4% Paraformaldehyde (PFA, 16005 Sigma Aldrich). The mouse brains were sectioned at 80 μm thickness and classic Nissl staining was performed to verify the locations of electrodes.

**Nissl staining**. Paraformaldehyde (PFA) fixed brains (4% in 0.1 M PBS) were washed in 0.1 M PBS, dehydrated in ethanol solutions with increasing concentrations (70, 80, 90, 96, 100%, three times, 30 min each), cleared in xylol (three times, 15 min and overnight), incubated (twice at 60 °C, overnight and 2 h), and finally embedded in paraffin. Sagittal brain slices (8 μm) were mounted on poly-L-lysine-coated glass slides. Paraffin was removed by incubation of the slices with RotI-histol ($2 \times 10$ min, Carl Roth GMBH) and subsequent treatment with ethanol 100, 96, and 70% (3 min, each) and rinsing in $H_2O$. Staining was performed in 0.1% cresyl violet solution for 5–10 min followed by short rinsing in $H_2O$ and two brief washes (in 96% ethanol). After dehydration in 100% ethanol (twice, 3 min each) slices were cleared in RotI-histol (twice, 3 min each) and mounted in a permanent mounting medium (Eukit, Sigma-Aldrich).

**Apoptosis (TUNEL) assay**. Mouse brains, perfused with PBS, were immersed in TissueTek (Sakura Finetek), rapidly frozen in liquid nitrogen and stored at –70 °C. The occurrence of apoptosis in 10-μm-thick brain sections was analyzed using the NeuroTacs II kit (Trevigen) according to the manufacturer's instructions. Importantly, with the exception of the terminal deoxynucleotidyl transferase TdT addition in some brain sections for the positive control, all slices were treated equally throughout the assay procedure. Sections were then counterstained with cresyl violet, dehydrated, and cleared in Histoclear (Thermo Fisher Scientific) before being coverslipped.

**Timm staining**. For Timm staining of hippocampal slices, we used the method of Denscher[108] with minor mofiications[106].

**Immunohistochemical (IHC) analysis**. Mice were anesthetized by isoflurane inhalation and transcardially perfused with ice-cold PBS (pH 7.5) and ice-cold 4% paraformaldehyde in PBS. Brains were post-fixed for 4 h at RT or overnight at 4 °C in the same fixative. For immunofluorescence (IF) assays, 50–70-μm-thick Vibrotome brain slices were placed in 24-well plates free-floating in PBS. All incubation steps were performed gently shaking. Slices were blocked for 2 h at RT in blocking buffer (5% normal goat serum, 0.5% Triton X-100, 1% bovine serum albumin in PBS) followed by 5 min wash in 0.5% Triton X-100 in PBS[73,105,106,109]. Primary antibodies were applied in 0.5% Triton X-100, and 1% bovine serum albumin in PBS at 4 °C. Slices were washed $3 \times 10$ min in 0.5% Triton X-100 in PBS before secondary antibody (dilutions see below) was applied and incubated for 2 h at RT. Slices were washed in PBS $2 \times 10$ min at RT and counterstained with DAPI diluted 1:10,000 in PBS for 15 min at RT. After a final wash in PBS, slices were mounted on glass slides (Menzel-Gläser), air-dried for 10 min, and embedded in aqua polymount (Polyscience). Antibodies used: NeuN (1:1000, MAB337; Millipore; RRID: AB_2313673), anti-GFAP (1:400, Z0334; AgilantDako; RRID: AB_10013382), anti-Calbindin (1:3000, CB38a; Swant; RRID:AB_10000347), anti-Parvalbunmin (1:3000, P3088; Sigma; RRID:AB_477329), anti-c-Fos (1:10000, (Ab-5)(4–7) Rabbit pAb, PC05; Millipore and 1:1000; Merk Millipore ABE457; RRID: AB_2631318), and anti-ARC (p16-Arc; 1:1000; sc-166760; Santa Cruz Biotechnology; RRID:AB_2060065) were used as primary antibodies. Secondary anti-mouse, either Cy3- (RRID:AB_2338683) or FITC-coupled (RRID:AB_2338594), and anti-rabbit, either Cy3- (RRID:AB_2338000) or FITC-coupled antibodies (1:300 each, all purchased from Jackson Immuno Research Labs, RRID: AB_2337972), were used for IF. Secondary anti-mouse (RRID:AB_2336176) and anti-rabbit antibodies coupled to horseradish peroxidase (1:600, Vector; RRID: AB_2336176) were used for staining with diaminobenzidine (DAB). DAPI (1:5000) was added in some IF experiments before the final washing steps with PBS.

**Nest building and burrowing**. For the analysis of nesting[37] and burrowing[36] behavior, mice were individually housed in the home cage and had free access to food and water. First, a piece of paper tissue was placed in the home cage. On the following day, the appearance of the nest was scored as follows: 1 = paper is untouched; 2 = paper is partially torn up; 3 = paper is shredded but there is no identifiable nest site; 4 = flat nest; 5 = perfect nest. The test was done for three consecutive days. For burrowing, 2.5 h before the start of the dark cycle a burrowing tube (L: 20 cm; Ø: 7 cm) filled with 200 g of food pellets was placed into the

home cage. Both at 2 and 18 h later the remaining food pellets in the burrowing tube were weighed to determine the burrowing activity.

**Assessment of clasping reflex**. For the assessment of the clasping reflex, the mouse was suspended by the end of its tail and elevated (10 cm) from the ground for 1 min. The behavior of paw- and limb-clasping was monitored[39].

**Acoustic stimulation and systemic drug administration**. Mice were placed in an acrylic glass chamber ($46 \times 24 \times 23$ cm) with sawdust on the floor. The chamber could be divided into two or four equal-sized compartments, which allowed the simultaneous acoustic stimulation of different genotypes. Acoustic stimulation (11 kHz, 110 dB) was delivered through speakers located on the top, center of the chamber. The stimulus was presented until the onset of AGS or for a maximum of 80 s ($4 \times 20$ s interrupted by 2-s breaks). Seizure responses were recorded by a webcam. Pharmacological treatment with the NMDAR antagonists memantine (5 mg/kg; M9292 Sigma) or dizocilpine (MK-801: 0.2 mg/kg; Cat. No. 0924 Tocris), or vehicle (PBS or saline) was by i.p administration. Drugs were injected 2–3 h before the first acoustic stimulation. If the AGS response was prevented the protocol was repeated without any further pharmacological intervention after 24, 48, and 120 h.

**MK-801-induced c-Fos expression**. To assess drug-induced c-Fos expression, mice were i.p.-injected with [+]-5-methyl-10,11-dihydro-5H-dibenzo-[a,d]-cyclo-hepten-5,10-imine hydrogen maleate (MK-801: 0.5 mg/kg, Cat. No. 0924 Tocris). All homozygous $Grin2a^{S/S}$ mice injected with MK-801 exhibited similar hyperactivity and disturbed motion as $Grin2a^{+/+}$ controls[110]. Animals were killed 2 h after treatment and the brains were prepared for IHC analysis.

**Quantification of c-Fos expression**. Bright-field images of DAB-stained slices were taken with the Axio imager 2 (Zeiss), which was connected to an Axiocam (Zeiss). Images were acquired at a resolution of 4000–6000 pixels per inch and saved by Axiovision (Zeiss). With ImageJ (NIH) respective brain regions were selected manually and the c-Fos-positive cells within these areas were manually counted by an observer blind with respect to genotype.

**In vivo local field potential oscillations in the hippocampus**. For in vivo neuronal network oscillations in the dorsal hippocampus of freely moving mice during their night cycle, two octrodes (eight twisted wires) were implanted into the right and left dorsal CA1 subregions[111]. One week after surgery, recordings were performed for 5 h in the animal's home cage. Extracellular signals were filtered (1–500 Hz), amplified (RHA2116, Intan Technologies), digitized (2.5 kHz), and stored for offline analyses with custom-written MATLAB routines (MathWorks). We restricted our analyses to selected REM sleep phases and awake exploratory phases, where both characteristic and robust theta-nested gamma oscillations (4–12 and 30–140 Hz) occur. REM sleep was identified by the absence of movement on all accelerometers accompanied by prominent regular theta oscillations while exploration showed similar theta oscillation but exhibited prominent movement signals on the accelerometers. Power spectral density analysis was calculated by means of the Welch periodogram method using the pwelch function from the "Signal Processing Toolbox" of MATLAB (50% overlapping, 4-s Hamming windows). To measure the intensity of coupling between the phase of a slow oscillation and the amplitude of a fast oscillation, we computed the MI (for details see ref. [66]). In brief, the MI is a measure to evaluate the divergence of phase–amplitude coupling from a uniform distribution, normalized such that values range between 0 (no coupling) and 1 (maximal coupling). We next selected the MI peaks to evaluate the maximum MI values of low gamma and high gamma to theta coupling, respectively, for further statistical analyses.

**Behavioral analysis**. Before the behavioral experiments, mice were familiarized to the experimenter (who was blind to the genotypes) with repeated handling. Each mouse was handled for at least two sessions per day for 3–5 days (2–3 min per session) by the experimenter wearing the same lab coat for the entire behavioral test block. For the familiarization, the mouse was taken out of its home cage and was allowed to explore the experimenter's arms and pockets in the lab coat and to get familiar with an open field before being moved back to its home cage. Embedding of the home cage was exchanged by the experimenter once a week. The behavioral experiments were performed during the light cycle between 9 a.m. to 5 p.m. For each experimental day, the mice were relocated in their home cages to the experimental room 1 h before the first experiment. For the operant olfactory conditioning and for appetitively motivated maze tasks, mice were assigned to a restricted water or food diet respectively, aimed at keeping the mice at 85–90% of their free-feeding weight. Food was supplied ad libitum to water-deprived mice and water to food-deprived animals. Experiments were conducted with several counterbalanced, gender- and age-matched mouse cohorts (3–10 months of age). During regular housing conditions, $Grin2a^{S/S}$ and $Grin2a^{+/S}$ mice showed no increased mortality. In 3 years of monitoring, the $Grin2a^{S/S}$ mice were born close to the expected Mendelian ratio from $Grin2a^{+/S} \times Grin2a^{+/S}$ matings ($Grin2a^{S/S}$ 20.53%, $Grin2a^{+/S}$ 54.0%, $Grin2a^{+/+}$ 25.47%; 263 animals total) and reached adulthood (6–12 months). $Grin2a^{+/S}$ and $Grin2a^{S/S}$ mice never showed any

spontaneous seizure episodes, neither in regular housing conditions nor during stressful behavioral tests and food or water restrictions or after local or oversea shipments. Similarly, *Grin2a^{S/S}* mice did not express any epileptic-like activity when exposed to flashing light or during in vivo LTP recordings in EEG recordings.

**Spontaneous locomotor activity**. Spontaneous locomotion of an individual mouse was measured by the number of crossings of light beams in 2 h in automated photocell transparent plastic activity cages ($26 \times 16 \times 17$ cm), during the light phase of the regular day/night cycle[112]. A thin layer of bedding (0.5 cm) was provided on the floor of each cage to reduce anxiety and encourage exploratory behavior. Mice had no prior exposure to the testing environment. Beam brakes were recorded in 5 min time bins.

**Open-field exploration with five novel objects**. Mice were first habituated to an open field ($60 \times 60 \times 30$ cm) in a 6-min test session. Then five different novel objects were placed in the middle and four outer quadrants of the open-field arena for a further 6-min exploration session. This object exploration session was repeated two more times with a 4-min inter-trial interval (ITI). Between these four sessions, the arena was not cleaned. Both the arena and objects were cleaned with 70% ethanol after an individual mouse completed all four sessions. The path length traveled by the animal was recorded and analyzed using the VideoMot2 system V6.01 (TSE Systems).

**CAR test**. For each mouse, the behavior was assessed on an elevated round plastic platform (50 cm above the ground; Ø, 20 cm). A CAR score[69] was determined in a 60-min session. The latency to the first fall from the platform was recorded for each mouse. Mice that fell off the platform were immediately put back on the platform and the total number of falls during the entire session was counted.

**Three-chamber social test**. Sociability was evaluated in an acrylic glass three-chamber box[113]. The box contained three equal-sized compartments (*W*: $25 \times L$: 16.7 cm): a central start chamber and left and right goal compartment. The central compartment was connected to the goal compartments through open doors ($3 \times 3$ cm). In the center of each goal compartment, a perforated acrylic glass cylinder (Ø, 10 cm) was located. In the first 10 min session, the mouse was habituated to the whole arena. In the second 10 min session, an age- and gender-matched C57Bl6/N "stranger" mouse was placed in the cylinder of one goal compartment and a non-living object (pencil sharpener) into the cylinder of the other goal compartment. The time spent and the distance traveled in each compartment were automatically recorded. The numbers of interactions were analyzed manually. Interactions were classified as all attempts to explore the cylinders encaging either an unfamiliar mouse or the novel object. A perimeter of 1.5 cm around the cylinder determined the threshold area for positive interactions. An interaction was counted positive when the mouse's nose had crossed the invisible boundary line around the cylinder or the mouse touched the cylinder with its nose. The mouse-to-mouse interactions vs. total interactions (mouse plus object combined) were calculated as a sociability score (i.e. a preference score for the social stimulus compared to the non-social object).

**Novel object recognition**. For novel object recognition, mice were first habituated to an open-field box made of black wood ($60 \times 60 \times 40$ cm) for 10 min a day on three consecutive days. On day 4, the box contained either two blocks or two bottles (e.g. A1 and A2). For this sample trial, the subject was given 10 min to explore the two identical objects. The subject was then returned to its home cage and after a 2 min interval, placed in the same box, which now contained one block and one bottle for a test trial (e.g. A3 and B1). Objects from the sample trial were never re-used for the test trial. Instead, another identical copy of the object was used. The identity of the object presented on the sample trial (A or B), as well as the location of the novel object in the test trial, was counterbalanced with respect to genotype. The box and the objects were cleaned with 70% alcohol after each trial. An experimenter who was blind with respect to the genotype of the mice recorded each trial and then manually scored the number of interactions the mouse had with the objects.

**Spatial novelty preference test in a Perspex Y-maze**. Mice were tested in a clear Perspex Y-maze with a white plastic floor, which was covered with a thin layer of sawdust[114]. On the sample run, the mouse could explore two open arms in the Y-maze for 5 min. After an inter-trial period of 1 min, the 2-min test run was performed in the same Y-maze with now all three arms open. In both the sample and test trials the time spent in each arm and the numbers of entries were manually counted.

**Operant olfactory conditioning**. Mice were assessed on a two-odor sequential discrimination task using semi-automated olfactometers (KNOSYS)[109,115]. In brief, mice were trained either to lick during the exposure to a positive conditioned odor stimulus (OS+) to retrieve a 2–4 μl water droplet (reward) or not to lick during a not rewarded odor (OS−) to avoid a 5 s increase in the 2 s ITI ("an additional time out punishment for a wrong response"). Licking was determined by a sensitive,

ohmic lick sensor at the water port. After pre-training mice were trained to discriminate between 1% amyl acetate (AA) and 1% ethyl butyrate (EB) over 500 trials using a go/no go-paradigm (250 trials for each stimulus, pseudo-randomized in 20-trial blocks, and counterbalanced for the identity of the rewarded stimulus across animals; all odors were dissolved in mineral oil).

**T-Maze**

*Apparatus*. Learning was assessed in an elevated wooden T-maze, consisting of a start arm ($47 \times 10$ cm) and two identical goal arms ($35 \times 10$ cm), surrounded by a 10-cm-high wall. A metal food well was located 3 cm from the end of each goal arm. The maze was located 1 m above the floor in a well-lit laboratory that contained various prominent distal extramaze cues.

*Spatial working memory*. Spatial working memory was assessed using a rewarded alternation (non-matching to place) paradigm in the T-maze[72,75,76,116]. In this test, each trial consisted of a sample run and a choice run. On the sample run, the mice were forced either left or right by the presence of a wooden block, according to a pseudorandom sequence (with equal numbers of left and right turns per session, and with no more than two consecutive turns in the same direction). The reward was available in the food well at the end of the arm. The mouse was then replaced in the start arm, facing away from the junction, with the block removed, allowing a free choice of either arm. The time interval between the sample run and the choice run was approximately 15 s. The animal was then rewarded for choosing the previously unvisited arm (i.e. for alternating). Mice were run one trial at a time with a minimum ITI of approximately 10 min. Mice received 30 trials in total.

*Visuo-tactile discriminations*. For the visuo-tactile discriminations floor inserts were placed as intra-maze cues into the T-maze according to the required experimental protocol[91]. The animals were required to use the floor insert information (e.g. black foam versus light blue toweling, white Perspex versus gray wire mesh) in order to guide their choices and find the food reward in either the left or the right goal arm of the T-maze apparatus.

**Simple cue discrimination task**. For the simple cue discrimination task, each goal arm contained a different floor insert with an attached food well, positioned 2 cm from the end of each insert. The milk reward was always placed on a specific floor insert (i.e. foam or towel) for equal numbers of animals of each genotype; this relationship remained constant throughout acquisition. Before testing, mice received one session of forced trials with only one of the goal arms available at a time to expose them to both floor inserts and their contingencies. This session comprised six rewarded and six non-rewarded trials in a pseudorandom sequence. The animal then received training sessions during which it was allowed a free choice of either arm and was rewarded for choosing the correct insert. Entry into an arm was defined when a mouse placed all four paws into that arm. Mice received six sessions comprising 10 trials per session with an ITI of approx. 15–20 min. The spatial orientation of the rewarded/non-rewarded arm (left versus right goal arm) varied from trial to trial in a pseudorandom sequence. During session 6, the milk reward was placed into the food well only after the mouse had chosen the arm to ensure that the mice were unable to solve the task by smelling the reward. Following the acquisition of the task, the mice were then trained on a reversal during which the cue/reward contingencies were reversed (i.e. mice trained on A+/B− now experience A−/B+; Supplementary Methods).

**Contiguous task**. Mice were trained on a conditional learning task with the floor inserts (white Perspex versus gray wire mesh) extending throughout the entire maze, including the start arm and both of the goal arms. For half of the animals, the presence of the Perspex insert indicated that the 0.1 ml milk reward was available in the left-hand goal arm. In contrast, the reward was in the right-hand goal arm if the maze contained the wire mesh floor insert. For the remaining mice, the opposite pair of floor insert/reward contingencies applied (e.g. Perspex/right, wire mesh/left). The relationship between the floor insert and the rewarded goal arm was constant for each animal throughout the experiment. Mice received 19 test sessions comprising 10 trials per session with an ITI of 5–10 min. Each session consisted of five trials with each of the two floor inserts, and no more than three consecutive trials with the same floor insert, according to a pseudorandom sequence. During session 19 the milk reward was delivered into the food well only after the mouse had made a choice. This was to ensure that the mice were unable to solve the task by smelling the milk reward.

**Discontiguous task**. Separate groups of mice were tested as described for the contiguous task above but with the only exception being that the floor inserts now covered just the start arm ($57 \times 10$ cm; extending right across to the wall opposite the start arm at the junction of the maze). Mice received 19 test sessions of 10 trials. During session 19 the milk reward was delivered into the food well only after the mouse had made a choice.

**Spatial reference memory**. Hippocampus-dependent SRM was assessed in the open-field watermaze using the same equipment and in the same laboratory with

the same set of spatial cues[73]. In order to escape from the water in an open-field watermaze (diameter 2.0 m) the mice had to find a fixed location, hidden escape platform (diameter 20 cm) submerged approximately 1 cm below the water surface. The platform was located at the center of either the NE, NW, SE, or SW quadrant of the pool. The number of mice trained to each platform position was counterbalanced with respect to the group. Animals had no swim pre-training prior to the start of spatial testing in the watermaze. They received four trials per day for 6 days, with an ITI of approximately 15 s. Mice were placed into the pool facing the side wall at one of eight start locations (nominally N, S, E, W, NE, NW, SE, and SW; chosen randomly across trials), and allowed to swim until they found the platform, or for a maximum of 90 s. Any mouse that failed to find the platform within the allotted time was lifted out of the water by the experimenter and placed onto the platform. The animal then remained on the platform for 30 s before commencing the 15 s ITI prior to the next trial. On the fourth (24 h after spatial training trial 12) and the seventh days of testing (24 h after spatial training trial 24), a probe trial was conducted to determine the extent to which the mice had learned about the spatial location of the platform. The platform was removed from the pool and the mice allowed to swim freely for 60 s.

**Radial maze and spontaneous alternation in the T-maze**. These methods are provided in the Supplementary Information.

## Data availability

Essential source data 1–11 were uploaded with the manuscript and can be downloaded from the internet links connected to this article Bertocchi et al., 2021 "Voltage-independent GluN2A-type NMDA receptor Ca$^{2+}$ signaling promotes audiogenic seizures, attentional and cognitive deficits in mice". Additional source data are available on request.

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

## Acknowledgements
We thank Bert Sakmann promoting and initiating this study, which is based on his in vitro recordings of modified recombinant modified NMDARs in his previous Max Planck Department in Heidelberg. We thank Annette Herold for expert technical assistance and genotyping of the mice. Our sincere thanks go to our colleagues Dr. Øivind Hvalby and Dr. Peter H. Seeburg, who were substantially involved in these experiments. Much to our deep regret, Øivind and Peter died far too early. We thank Andreas Draguhn for his support and critical comments on the manuscript. Part of this work were released in poster presentations at the Society of Neuroscience: Single et al. (2002) Meeting SfN Poster 150.6; Niewoehner et al. (2005) Meeting SfN Poster 611.1, Rawlins et al. (2008) at the FENS Forum Abstr. vol. 4, 157.23, 2008, at the British Neuroscience Association Meeting "Festival of Neuroscience" in Edinburgh 2015 Poster P2-D-049 and in the Ph.D. theses from B. Niewoehner (University of Oxford, 2005), from V. Pawlak (Heidelberg University, 2004), the bachelor thesis from T. Boerner

(Osnabrück University, 2010), the master theses from M. Serafino (University of Pisa, 2014), from M. Briese (University of Oxford, 2004), and the MD thesis of B. Yang (Heidelberg University, 2016). This work was supported in part by a grant from the Wellcome trust SRF 074385/Z/04/Z to D.M.B. and by the German Research council to R.S. (SP602/1, SFB636/A4, and SFB1134/B01). R.S. receives support from the Ingeborg Ständer Foundation. V.J. was supported by EU-Grant QLRT-1999-01022 LTP expression. V.N.C. received support by a postdoc fellowship from the German Research Foundation (NG 167/1-1). A.R. is supported by the "5–100" competitive growth program of Kazan Federal University and by the subsidy allocated to Kazan Federal University for the state assignment in the sphere of scientific activities.

## Author contributions
F.N.S. and R.S. generated and molecularly analyzed the mouse lines. V.J., V.P., A.R., and N.B. performed and analyzed the electrophysiological experiments in acute brain slices. S.-B.L. and G.K. performed in vivo LTP. recordings. V.N.C. and M. Both recorded and analyzed the hippocampal network activity. I.B., M.S., and H.A.O. carried out and analyzed the AGS. induction experiments. B.Y. and P.G. induced and anaylsed the c-Fos expression by MK-801 injections and H.S. quantified the signals. I.B., M.S., A.E. and T. Bus performed and analyzed the behavioral performance of mice in Heidelberg. B.N., T. Boerner, M. Briese, D.M.B., and J.N.P.R. performed and analyzed the behavioral performance of mice in Oxford. I.B., M.S., B.N., and T. Bus provided histological and immunohistological analysis. R.S., I.B., and D.B. wrote the manuscript.

## Funding

## Competing interests
The authors declare no competing interests.

## Additional information

Ilaria Bertocchi [1,2,3,4], Ahmed Eltokhi [2,5], Andrey Rozov [1,6,7,8,22], Vivan Nguyễn Chi [6,22], Vidar Jensen [9,22], Thorsten Bus [1,2], Verena Pawlak [1,10], Marta Serafino [1,11], Hannah Sonntag [1,2], Boyi Yang [1,12], Nail Burnashev [1,13], Shi-Bin Li [1,14,15], Horst A. Obenhaus [1,2,16], Martin Both [6], Burkhard Niewoehner [17], Frank N. Single [1,18],

Michael Briese[17,19], Thomas Boerner[17], Peter Gass[20], John Nick P. Rawlins[17], Georg Köhr [1,21], David M. Bannerman [17✉] & Rolf Sprengel [1,2✉]

[1]Departments Molecular Neurobiology and Physiology at the Max Planck Institute for Medical Research, Jahnstr. 29, 69120 Heidelberg, Germany. [2]Research Group of the Max Planck Institute for Medical Research at the Institute for Anatomy and Cell Biology of the Heidelberg University, Im Neuenheimer Feld 307, 69120 Heidelberg, Germany. [3]Department of Neuroscience Rita Levi Montalcini, University of Turin, Via Cherasco 15, 10126 Torino, Italy. [4]Neuroscience Institute–Cavalieri-Ottolenghi Foundation (NICO), Laboratory of Neuropsychopharmacology, Regionale Gonzole 10, 10043 Orbassano, Torino, Italy. [5]Department of Neurology and Epileptology, Hertie Institute for Clinical Brain Research, Eberhard Karls University Tübingen, Otfried-Müller Str. 27, 72076 Tübingen, Germany. [6]Department of Physiology and Pathophysiology, Heidelberg University, Im Neuenheimer Feld 326, 69120 Heidelberg, Germany. [7]OpenLab of Neurobiology, Kazan Federal University, 8 Kremlyovskaya Street, Kazan 420008, Russian Federation. [8]Federal Center of Brain Research and Neurotechnologies, Ostrovityanova Str 1/10, Moscow 117997, Russia. [9]Department of Molecular Medicine, Division of Physiology, Institute of Basic Medical Sciences, University of Oslo, Sognsvannsveien 9, 0372 Oslo, Norway. [10]Department of Behavior and Brain Organization, Research Center Caesar, Ludwig-Erhard-Allee 2, 53175 Bonn, Germany. [11]FARMA-DERMA s.r.l. Via dell'Artigiano 6-8, 40010 Sala Bolognese, Italy. [12]Department of Geriatrics, Tongji Hospital, Tongji Medical College, Huazhong University of Science and Technology, 1095 JieFang Road, Wuhan, Hubei 430030, China. [13]INSERM UMR 1249 Mediterranean Institute of Neurobiology (INMED), Aix-Marseille University, Parc Scientifique de Luminy, 163 avenue de Luminy BP13, 13273 Marseille Cedex 09, France. [14]Department of Psychiatry and Behavioral Sciences, Stanford University School of Medicine, 1201 Welch Road, Stanford, CA 94305, USA. [15]Wu Tsai Neurosciences Institute, Stanford University, Stanford Way, Rm E152, Stanford, CA 94305, USA. [16]Kavli Institute for Systems Neuroscience, Faculty of Medicine and Health Sciences, NTNU, Postboks 8905, NO-7491 Trondheim, Norway. [17]Department of Experimental Psychology, University of Oxford, Radcliffe Observatory, Anna Watts Building, Woodstock Rd, Oxford OX2 6GG, UK. [18]Miltenyi Biotec B.V. & Co. KG, Bergisch Gladbach, Friedrich-Ebert-Str. 68, 51429 Bergisch Gladbach, Germany. [19]Institute of Clinical Neurobiology, University Hospital Wuerzburg, Versbacherstraße 5, 97080 Wuerzburg, Germany. [20]RG Animal Models in Psychiatry, Animal Models Psychatry, Central Institute of Mental Health (CIMH), Faculty Mannheim, Heidelberg University, J5, 68159 Mannheim, Germany. [21]Department of Neurophysiology, Medical Faculty Mannheim, Heidelberg University, Ludolf-Krehl-Str. 13–17, 68167 Mannheim, Germany. [22]These authors contributed equally: Andrey Rozov, Vivan Nguyễn Chi, Vidar Jensen. ✉email: Rolf.Sprengel@mpimf-heidelberg.mpg.de; David.Bannerman@psy.ox.ac.uk

