## [Peer Review File · Communications Biology]

Reviewers' comments:

Reviewer #1 (Remarks to the Author):

This manuscript describes in detail the evaluation of phenotypes in transgenic mice with GluN2A subunits containing the N615S mutation. This mutation is known to reduce Mg²⁺ block of GluN2A-containing NMDA receptors with little effect of Ca²⁺ permeability. The authors describe many novel findings that are important to our understanding of GluN2A-containing NMDA receptors in brain function as well as pathologies associated with de novo mutations in the GRIN2A gene. The transgenic mice described in the paper will likely provide an important new tool for future studies. The interesting phenotypes described for these transgenic mice nicely complement the phenotypes found in GluN2A KO mice, and the authors do an excellent job at integrating their findings with the body of literature in the field. The experiments are carefully constructed and technically challenging. The manuscript is extremely well written, the conclusions drawn are well substantiated, the data is nicely presented, and the figures are high-quality. Overall, this is an impressive and important study that addresses a timely and clinically relevant area of research. I only have very few points that the authors may want to consider.

- 1) Lines 103 and 634: "GLUN2A" should be "GluN2A"
- 2) Line 116: "NMDR" should be "NMDAR"
- 3) Fig. 1e,f: The author should state the age of the mice used for these experiments. In supplemental, please define the age of "adult" mice.
- 4) Lines 688-689: The authors should state the dilution and source of all antibodies used.

Reviewer #2 (Remarks to the Author):

The manuscript by Bertocchi and colleagues presents some interesting findings that help to tease out the role of the activity-dependent aspect of GluN2A-containing NMDA receptors on brain function. The study is well constructed and is able to go beyond a series of phenotype descriptions. I enjoyed the paper and only have minor comments.

- 1) Abstract, line 108, maybe place a comma after "hippocampus" for clarity
- 2) 116, typo NMDR
- Line 116-117: "NMDR Ca²⁺ signaling is necessary for homeostatic processes that regulate electrical activity and cognitive behavior." Not exactly sure what this tells me, this is a critical sentence for the paper that could increase the paper's impact. I leave it to the authors as they have thought about this much more, but maybe one could emphasize the activity-dependency or voltage-dependency (rather than the calcium signaling which is unchanged) of GluN2A-containing receptors and state that this is necessary for specific forms of associative learning and cognition as theoretically predicted for a coincidence detector. Hard to summarize this concept with seizures in the same sentence.
- 3) Line 155. Ref. 17 doesn't establish that GluN1/GluN2s are the most abundant, the paper was published before the cloning of GluN3s.
- 4) Line 163, Ref. 22 concludes that GluN2B-KO neonates do not die from respiratory failure like GluN1-KOs.
- 5) Line 335, "significant increase" could potentially be interpreted as significant to un-injected 2A-KO mice or significant compared to injected WT mice.
- 6) Line 536, also 573. Maybe there is a reason, but the authors refer to "unregulated" and "uncontrolled" GluN2A receptors. The receptors are still ligand-gated, so would it be better to refer to "voltage-independent" or "activity-independent"?
- 7) Line 604, "unregulated", does that mean "non-habituating" in this context or is that inferring

too much?

8) Line 346, "as shown above (Fig. 3e)", the figure might not end up being place above.

9) Fig. 2f "IML" labelling appears to be pointing to the granule cell layer, not the inner molecular layer.

10) Fig. 1d, It is difficult to distinguish the color of the traces for AMPARs and NMDARs.

11) Fig. 3a, c, e. These have 3 different Y-axis labels, but are they different? The legend says 3a is mortality, but the axis label says it is % AGS (audiogenic seizure). Axis label for 3c indicates AGS-induced respiratory arrest, while 3e is just "AGS in %", but the legend says that they are the same with just memantine vs MK-801 being the difference. Might distinguish between AGS and AGS-RA in some consistent manner.

12) Fig. 3a, the s/s mouse appears to have two different values at the 2nd time point.

13) Fig. 5 legend, line 1024, indicates gamma oscillations as 150-200 Hz. The figure appears to show all frequencies between ~15 Hz and 200 Hz which includes gamma (and other frequency bands).

Reviewer #3 (Remarks to the Author):

In the manuscript entitled "Activity-regulated GluN2A-type NMDA receptor Ca²⁺ signaling differentially controls neuronal circuits involved in audiogenic seizures, attention and cognitive performance" Bertocchi and colleagues use a knock-in mouse model, GluN2A-N615S, to explore the impact of loss of voltage-dependent magnesium block on neurophysiology. The authors present exciting results, showing changes in circuit excitability, a host of behavioral assays, and synaptic plasticity. These results add another layer of complexity and depth to our understanding of NMDA receptor action on brain function. Thus, this manuscript will be of general interest to the field and readers of Communications Biology. However, one potentially exciting aspect of this manuscript, clinical translation, is weakened by the lack of data on human patients that host this particular missense variant and the absence of comprehensive discussion of similar human variants. Nevertheless, the manuscript is scientifically sound and an important addition to the field. There are several features of the text that could be improved, including presentation of raw data, more detail on statistics, and addition of detail to methods.

Major concerns

1. Brain slice methods should include a description of the composition of the cutting solution, slice thickness, Ca²⁺ concentration, recording temperature, acquisition rate, filtering rate, internal solution composition, etc. are all missing and are all important consideration in slice electrophysiological experiments.

2. Methods are also missing a description of statistical analysis. Are the experiments correctly powered, if so, to what level? What statistical tests were performed? How did the authors control for family-wise error when measuring more than one feature from an experiment?

3. All bar graphs should be modified to show individual data points so the reader can assess variability in the data.

Minor concerns:

4. The authors have not included reference to functional studies of multiple nearby residues that have been identified in patients with neurological disorders and are published (2A-N614S, 2A-N615K, 2B-N615I, 2B-N615K, 2B-N616K, see Strehlow et al., Li et al). I think this might be valuable for the reader.

5. Although the GluN2A-N615S variant is well characterized in terms of calcium permeation and magnesium block at different voltages, there are still multiple properties that have not been described in the literature. Agonist potency, deactivation, peak and steady-state amplitude, charge transfer, desensitization, efficacy of pore blockers (like memantine and MK-801 used in the study) and surface expression are unknown. Even though the aim of this paper was to perturb voltage-dependent block of GluN2A-containing NMDARs, changes in these other parameters might also impact neurophysiology. Thus, without an in vitro characterization of this mutation, it is difficult to conclude with certainty that this variant causes the described phenotypes due to loss of magnesium block alone. These are straightforward experiments, some of which should be added. Interestingly, the variant exists in human patients and some functional characterization has been performed, although apparently not published in the peer reviewed literature (<http://functionalvariants.emory.edu/database/index.html>). This kind of information would be a valuable addition and strengthen the overall conclusions.

6. It would be helpful to know whether the N-S mutation alters the IC50 or voltage dependence of memantine block. I wonder if actions of memantine reflect block of 2B subunits rather than 2A mutant subunits, which should be insensitive to memantine. Some discussion of this possibility seems appropriate.

7. The rationale behind the following conclusion is not well explained: "Together, these results show an incorporation of GluN2A(N615S) into synaptic NMDARs but the failure of GluN2A(N615S) receptors to participate in LTP." Perhaps clarify or expand the text?

8. The LTP work is well done, important for the manuscripts story, and should be part of figure 1.

9. The overall layout of the figures does not transparently illustrate the data or variability. For example, figure 1 shows representative bands for a western blot but no density measurements. Inside the figure legend, it states "t-test, $p > 0.05$ ". Where is the densitometry data, how many observations were made, etc. This issue is repeated in other figures.

10. Figure 2 legend doesn't mention whether quantification or any statistical test was performed. As written, it appears that this is a result of a single experiment, although I am certain the authors replicated the data an appropriate number of times. Perhaps expand discussion to indicate number of experiments performed, quantification, etc..

11. The GluA1 representative IHC images in Fig 2 look overexposed to a point where differences in GluA1 expression levels might be missed.

12. Manuscript has a lot of data that could be organized more clearly. Perhaps consider combining Figures 6 and 7 into one figure, with some portions going to supplemental? This is only a suggestion, and the authors should simply consider the best way to organize data.

13. Typo in abstract, accidentally say "NMDR" instead of "NMDAR"

14. Typo in abstract and in introduction where "GLUN2A" is in all capital letters

Response to the reviewers' comments:

We are thankful for the very constructive and supportive criticisms provided in the reviewer's comments. We have addressed all of their queries to improve the quality of the manuscript.

Reviewer #1 (Remarks to the Author):

This manuscript describes in detail the evaluation of phenotypes in transgenic mice with GluN2A subunits containing the N615S mutation. This mutation is known to reduce Mg²⁺ block of GluN2A-containing NMDA receptors with little effect of Ca²⁺ permeability. The authors describe many novel findings that are important to our understanding of GluN2A-containing NMDA receptors in brain function as well as pathologies associated with de novo mutations in the GRIN2A gene. The transgenic mice described in the paper will likely provide an important new tool for future studies. The interesting phenotypes described for these transgenic mice nicely complement the phenotypes found in GluN2A KO mice, and the authors do an excellent job at integrating their findings with the body of literature in the field. The experiments are carefully constructed and technically challenging. The manuscript is extremely well written, the conclusions drawn are well substantiated, the data is nicely presented, and the figures are high-quality. Overall, this is an impressive and important study that addresses a timely and clinically relevant area of research. I only have very few points that the authors may want to consider.

1) Lines 103 and 634: "GLUN2A" should be "GluN2A"

GLUN2A was changed to **GluN2A**

2) Line 116: "NMDR" should be "NMDAR"

"NMDR" was changed to **"NMDAR"**

3) Fig. 1e,f: *The author should state the age of the mice used for these experiments. In supplemental, please define the age of "adult" mice.*

The age of adult mice used for the field LTP experiments corresponding method section. In addition, the short description of the methods was replaced by a detailed description of the methods. A video (Supplementary data; Movie 1) of the slice preparation is now included in the **"Supporting Information"** and is available for downloads.

Hippocampal field LTP recordings in acute brain slices

Local field potential recordings at hippocampal CA3-to-CA1 synapses in transverse, acute brain slices were performed as previously described^{36,104-107}. In brief:

Slice preparation. Adult mice (2 – 4 months, and 3 – 5 mice, per genotype and experiment) were sacrificed with Suprane (Baxter) and the brains were gently removed from the skull. Transverse slices (400 μm) were cut from the middle and dorsal portion of each hippocampus (see **Supplementary data: Movie 1**) with a vibroslicer (Campden Instruments NVSLM1) in cold artificial cerebrospinal fluid (aCSF, 4°C, bubbled with 95% O₂ - 5% CO₂) containing (in mM): 124 NaCl, 2 KCl, 1.25 KH₂PO₄, 2 MgSO₄, 1 CaCl₂, 26 NaHCO₃ and 12 glucose. Slices were placed in an interface chamber exposed to humidified gas at 28 – 32°C and perfused with aCSF (pH 7.3) containing 2 mM CaCl₂ for at least 1h prior to the experiments. In some of the experiments, DL-2-amino-5-phosphopentanoic acid (AP5, 50 μM; Sigma-Aldrich) was added to the aCSF in order to block NMDAR-mediated synaptic plasticity or a 10 μM concentration of GluN2B-specific antagonist CP-101,606 (CP) (Pfizer) was added to the perfusion media.

Synaptic excitability. Orthodromic synaptic stimuli (<300 μ A, 0.1 Hz) were delivered through tungsten electrodes placed in either *stratum radiatum* proximal or *stratum oriens* distal of the hippocampal CA1 region. The presynaptic volley and the field excitatory postsynaptic potential (fEPSP) were recorded by a glass electrode (filled with ACSF) placed in the corresponding synaptic layer while another electrode placed in the pyramidal cell body layer (*stratum pyramidale*) monitored the population spike. Following a period of at least 10 – 15 min with stable responses, we stimulated the afferent fibers with increasing strength. A similar approach was used to elicit paired-pulse responses (50 ms interstimulus interval, the two stimuli being equal in strength). To assess synaptic transmission, we measured the amplitudes of the presynaptic volley and the fEPSP at different stimulation strengths. In order to pool data from the paired-pulse experiments we selected responses to a stimulation strength just below the threshold for eliciting a population spike on the second fEPSP.

Long-term potentiation of synaptic transmission. Orthodromic synaptic stimuli (50 μ s, < 300 μ A) were delivered alternately through two tungsten electrodes, one located in the *stratum radiatum* and another one in the *stratum oriens* of the hippocampal CA1 region. Extracellular synaptic potentials were monitored by two glass electrodes (filled with aCSF) placed in the corresponding synaptic layers. After obtaining stable synaptic responses in both pathways (0.1 Hz stimulation) for at least 10 – 15 minutes, one of the pathways was tetanized (either with a single 100 Hz tetanization for 1 sec, or repeated four times at 5 min intervals). To standardize the procedure, the stimulation strength used for tetanization was just above threshold for generating a population spike in response to a single test shock. Synaptic efficacy was assessed by measuring the slope of the fEPSP in the middle third of its rising phase. Six consecutive responses (1 min) were averaged and normalized to the mean value recorded 1 – 4 min prior to tetanization. Data of the same experimental group were pooled across animals and are presented as mean \pm SEM (see also³⁶). Statistical significance was evaluated by using a linear mixed model analysis (SAS 9.1) with $p < 0.05$ being designated as statistically significant.

Hippocampal field LTP recordings in freely moving mice: Local field potential recordings at hippocampal CA3-to-CA1 synapses in freely moving mice were performed as previously described¹⁰⁸. Briefly, 2 month old mice were deeply anaesthetized with a mixture of Ketamine (65 mg/kg) and Xylazine (14 mg/kg). In a stereotaxic frame, two mini-screws were fixed above the cerebellum serving as reference and ground electrodes respectively. Then, a bipolar stimulation electrode (two insulated tungsten wires glued together, each 52 μ m in diameter, California Fine Wire) and a recording electrode (single wire, same material with stimulation electrode) were positioned in *stratum radiatum* by a motorized manipulator (Luigs & Neumann) according to the criteria described previously¹⁰⁸. Electrodes were permanently fixed with dental acrylic and surgical wounds were sutured. Singly-housed mice were allowed to recover with access to food and water *ad libitum* for at least one week before recording.

Mice were put in the recording chamber (50 cm diameter round arena, 50 cm high) for environmental acclimation overnight. A miniature headstage (1 g, npi electronic GmbH, Tamm) was connected to electrodes/pins in the presence of 95% O₂ containing 4% isoflurane for stress relief. Evoked LFPs were filtered with a band width of 0.3 to 500 Hz, amplified with the miniature head stage (EXT-02F, npi electronic GmbH), and stored at 10 kHz (ITC-16, HEKA Elektronik) after 50 Hz noise filtration by a Hum Bug Noise Eliminator (AutoMate Scientific, Inc.). Extracellular stimulation was generated with an isolated stimulator (A365, WPI). The slopes of evoked LFPs were analyzed based on the middle one third of the rising phase (Fitmaster, HEKA Elektronik). At the beginning of each recording, two IO curves per mouse were generated, applying stimulation voltages with both polarities. To evaluate changes in synaptic efficacy, a stimulus strength eliciting 35-40% of maximum slope was used as the test pulse and was given every 30 sec. For LTP induction, two trains of high frequency stimulation (50 \times 100 Hz, 100 μ s pulse width, same intensity as test pulse) separated by 5 min were used.

After recordings, mice were deeply anesthetized and electrical lesions were induced twice (20 μ A, 10 s) for each single tungsten wire. Subsequently, mice were perfused with PBS followed by 4% PFA. The mouse brains were sectioned at 80 μ m thickness and classic Nissl staining was performed to verify the locations of electrodes.

4) Lines 688-689: *The authors should state the dilution and source of all antibodies used.*

We apologize that we had overlooked providing this detailed information on the antibodies in this part of the methods in the original submission. In addition to the RRID numbers we have now included a detailed method as to how the samples were prepared for the immunoblot.

Immunoblots: Total lysates from isolated forebrains and immunoblotting were performed as described previously¹⁰⁷. In brief, forebrains were placed in a glass pestle into ice and 1ml of cold homogenization buffer (0.32M sucrose, 25mM HEPES pH 7.4, 0.5mM EDTA, protease and phosphatase inhibitors) was added to

homogenize the tissue using 10 – 15 strokes, avoiding formation of bubbles. Then, the homogenate was centrifuged at 2000 rpm for 5 min at 4°C to remove pellet nuclear fraction (P1). The supernatant (S1 = total lysate) was centrifuged at 14.000 rpm for 30 min at 4°C to yield crude cytosol (S2) and crude membrane pellet (P2). Pellet 2 was resuspended in lysis buffer (1M TrisHCl pH 7.6, 5M NaCl, 1M KCl, 10% TritonX100, 10% Nonidet P40, 0.5 M EDTA, protease and phosphatase inhibitors). Protein concentration of the lysate was measured by Bradford protein assay. Protein samples (10 µg/lane) were separated by SDS/PAGE and analyzed by standard immunoblotting.

We included now:

anti-GluN1 (1:1000; Millipore AB9864R, RRID:AB_10807557)

anti-GluN2A (1:2000; rabbit)

anti-GluN2B (1:1000 mouse)

anti-PSD95 (1:2000; AB9708, Millipore, RRID:AB_11212529)

anti-P-CamKII (1:2000; MA1-047, Thermo Scientific, RRID:AB_325402)

anti GAPDH (1:5000, AB8245, Abcam, RRID:AB_2107448)

anti-beta actin (1:15000; clone AC-15, A5441, Sigma-Aldrich, RRID:AB_476744)

secondary goat anti-rabbit (RRID:AB_2336198) and goat anti-mouse (1:15000, Vector; RRID:AB_2336171)

Reviewer #2 (Remarks to the Author):

The manuscript by Bertocchi and colleagues presents some interesting findings that help to tease out the role of the activity-dependent aspect of GluN2A-containing NMDA receptors on brain function. The study is well constructed and is able to go beyond a series of phenotype descriptions. I enjoyed the paper and only have minor comments.

1) Abstract, line 108, maybe place a comma after “hippocampus” for clarity

The beginning of the sentence “In contrast, in the hippocampus MK801-induced ...” was changed to “**In contrast in the hippocampus, MK801-induced...**”

2) 116, typo NMDR

Typo is now fixed to **NMDAR**.

Line 116-117: “*NMDR Ca²⁺ signaling is necessary for homeostatic processes that regulate electrical activity and cognitive behavior.*” Not exactly sure what this tells me, this is a critical sentence for the paper that could increase the paper’s impact. I leave it to the authors as they have thought about this much more, but maybe one could emphasize the activity-dependency or voltage-dependency (rather than the calcium signaling which is unchanged) of GluN2A-containing receptors and state that this is necessary for specific forms of associative learning and cognition as theoretically predicted for a coincidence detector. Hard to summarize this concept with seizures in the same sentence.

We are very thankful to the reviewer for this important comment to emphasize the coincidence detection of GluN2- containing NMDARs as a main feature of those receptors.

We changed the title and rephrased the entire manuscript with the emphasize to focus on the physiological effects that are due to the voltage-independent Ca²⁺ influx of GLuN2A(N615S) containing receptors.

Therefore, we changed the original title of the manuscript:

“Activity-regulated GluN2A-type NMDA receptor Ca²⁺ signaling differentially controls neuronal circuits involved in audiogenic seizures, attention and cognitive performance” into “**Voltage-independent**

GluN2A-type NMDA receptor Ca²⁺ signaling promotes audiogenic seizures, attentional and cognitive deficits”

In the abstract we replaced the sentence: “Together, our findings provide experimental evidence that the tight activity-regulated Ca²⁺ signaling of GluN2A-type NMDARs is essential for maintaining appropriate responses to sensory stimuli, supporting the theoretical assumption that the NMDAR²⁺ signaling is necessary for homeostatic processes that regulate electrical activity and cognitive behavior” with the sentence : “Together, our findings provide experimental evidence that the inherent voltage-dependent Ca²⁺ signaling of NMDA receptors is essential for maintaining appropriate responses to sensory stimuli”

In the Introduction we added the sentence:

The NMDAR function as a coincidence detector is generally identified with the induction of long term potentiation (LTP), the dominant experimental model of synaptic plasticity (Collingridge, Kehl & McLennan 1983). The voltage-controlled Mg²⁺ block is essential for this activity-dependent NMDAR signaling (Nicoll & Malenka, 1995.). In recombinant GluN1/2A NMDARs the Mg²⁺ block is predominantly determined by the asparagine amino acid residue GluN2A(N615). In oocytes and HEK293 cells expressing recombinant GluN1/GluN2(N615S) heterodimeric receptors, the GluN2A(N615S) mutation led to a pronounced attenuation of the Mg²⁺ block, but with only minor 1.4 fold increased Ca²⁺ permeability (Wollmuth et al.1996). Notably, a similar Mg²⁺ block attenuating point mutation (c.1841A>G, p.Asn615Ser) at the identical position of the GluN2A subunit was found in two unrelated young female patients who suffered from epileptic seizures, intellectual disability (ID), moderate hypotonia and speech/language disorders (Endele et al., 2010; Allan et al., 2011).

3) Line 155. Ref. 17 doesn't establish that GluN1/GluN2s are the most abundant, the paper was published before the cloning of GluN3s.

Thank you for bringing this to our attention. We replaced Ref. 17 by more recent reviews: Ewald and Cline 2009; Paoletti P, Bellone C, Zhou Q, 2013.

4) Line 163, Ref. 22 concludes that GluN2B-KO neonates do not die from respiratory failure like GluN1-KOs.

We thank the reviewer for this critical point. We removed the reference 22 at the end of this sentence and included an exact description of reference 22 in order to emphasize the importance of the GluN2B-type NMDAR activity for the neuronal network formation. The following sentence was inserted:

The importance of precise NMDAR-signaling for the establishment of autonomic pattern activity in neuronal circuits is further emphasized by *Grin2b* knockout mice. In GluN2B-deficient pups, the trigeminal neuronal pattern formation is impaired and the pups starve to death within the first days after birth due to the lack of suckling responses(Kutsuwada et l., 1996)

5) Line 335, “significant increase” could potentially be interpreted as significant to un-injected 2A-KO mice or significant compared to injected WT mice.

We added at the end of the sentence: ...Compared to MK-801 treated wild-type controls.

6) Line 536, also 573. Maybe there is a reason, but the authors refer to “unregulated” and “uncontrolled” GluN2A receptors. The receptors are still ligand-gated, so would it be better to refer to “voltage-independent” or “activity-independent”?

We agree that our description was imprecise and have replaced both terms by “voltage-independent” or “activity-independent” in the entire manuscript

7) Line 604, “unregulated”, does that mean “non-habituating” in this context or is that inferring too much?

We have changed the term “unregulated” to “dysregulated” throughout the manuscript. The referee is correct to note that from the data provided in the manuscript there is little if any evidence that these mice can habituate at all. However, to make the claim that these mice are always “non-habituating” would require additional experiments to assess other associative forms of long-term habituation which is beyond the scope of this current study.

8) Line 346, “as shown above (Fig. 3e)”, the figure might not end up being place above.

We moved the reference to Fig.3e to the end of the sentence (see Fig. 3e)

9) Fig. 2f “IML” labelling appears to be pointing to the granule cell layer, not the inner molecular layer.

We apologize for the wrong labelling of the DG layers and have relabeled Fig. 2f (N.B. in the revised version it is Fig. 3f).

10) Fig. 1d, It is difficult to distinguish the color of the traces for AMPARs and NMDARs.

In the new Figure 2 the traces for NMDAR currents are now given in green color

11) Fig. 3a, c, e. These have 3 different Y-axis labels, but are they different? The legend says 3a is mortality, but the axis label says it is % AGS (audiogenic seizure). Axis label for 3c indicates AGS-induced respiratory arrest, while 3e is just “AGS in %”, but the legend says that they are the same with just memantine vs MK-801 being the difference. Might distinguish between AGS and AGS-RA in some consistent manner.

We thank the reviewer for bringing this inconsistency to our attention.

We have re-labelled the y-axes of the figure now consistently for Fig. 3a,c, (Now Fig. 4) and with AGS-induced respiratory arrest.

12) Fig. 3a, the s/s mouse appears to have two different values at the 2nd time point. By accident the straight line had an angle.

We apologize for this graphical error. This is now fixed in the revised version of the figure (now Figure 4).

13) Fig. 5 legend, line 1024, indicates gamma oscillations as 150-200 Hz. The figure appears to show all frequencies between ~15 Hz and 200 Hz which includes gamma (and other frequency bands).

We have now fixed the typo (150 Hz was changed to 15 Hz) in the figure legend (now Fig. 6).

Reviewer #3 (Remarks to the Author):

In the manuscript entitled "Activity-regulated GluN2A-type NMDA receptor Ca²⁺ signaling differentially controls neuronal circuits involved in audiogenic seizures, attention and cognitive performance" Bertocchi and colleagues use a knock-in mouse model, GluN2A-N615S, to explore the impact of loss of voltage-dependent magnesium block on neurophysiology. The authors present exciting results, showing changes in circuit excitability, a host of behavioral assays, and synaptic plasticity. These results add another layer of complexity and depth to our understanding of NMDA receptor action on brain function. Thus, this manuscript will be of general interest to the field and readers of Communications Biology. However, one potentially exciting aspect of this manuscript, clinical translation, is weakened by the lack of data on human patients that host this particular missense variant and the absence of comprehensive discussion of similar human variants. Nevertheless, the manuscript is scientifically sound and an important addition to the field. There are several features of the text that could be improved, including presentation of raw data, more detail on statistics, and addition of detail to methods.

Major concerns

1. Brain slice methods should include a description of the composition of the cutting solution, slice thickness, Ca²⁺ concentration, recording temperature, acquisition rate, filtering rate, internal solution composition, etc. are all missing and are all important consideration in slice electrophysiological experiments. To add in material and methods.

In the revised version we have now added all of the important information requested to the Materials and Methods Section. In addition, we included a video demonstrating the preparation of the hippocampal slices used for *in vitro* field recordings (Supplementary Data; Movie 1).

The following methods sections were now extended (See response to reviewer 1).

Single cell electrophysiology

Hippocampal field LTP recordings in acute brain slices

Hippocampal field LTP recordings in freely moving mice

Supplementary Movie 3: Hippocampal Slice preparation

2. Methods are also missing a description of statistical analysis. Are the experiments correctly powered, if so, to what level? What statistical tests were performed? How did the authors control for family-wise error when measuring more than one feature from an experiment? ??? maybe he wants the eta-squared?

In the METHODS we have now included the section: *Statistics and reproducibility*.

Statistics and reproducibility

In all figures the number of independently recorded values is clearly indicated. The number of animals and the number of recorded data points are given as appropriate. In bar graphs all data points used are pictured together with the standard error of the mean. In the figure legends the tests used for statistical evaluation (ANOVA, t-test etc.) are stated together with the P-values of the results. P-Values indicating a significant difference are given directly in the figures. Due to space limitations in the main figure legends, detailed descriptions of the statistical analyses can be found in the Supplementary Information: Supplementary Statistics to Figs. 1, 2, 3, 4, 7 and 8. In the Supplementary Figures the

details of the statistical analyses are given in the figure legends. When multiple comparisons were used to control the familywise error rate, we indicate the statistical test used (e.g. Bonferroni test). When appropriate, non-parametric analyses (e.g. Mann-Whitney U-tests) were conducted. Data records from automated behavioral analyses can be provided on request.

In addition, the statistical values are given in the figure legends. For reasons of space-limitation we have moved the description of the statistics of the main figures to the supplementary material. For the supplementary figures the statistics are given directly in the supplementary figure legend.

-The odor relearning experiment was removed due to the low number of animals.

-The body weight graph was replaced by a graph describing animals of very similar age raised in one cohort.

-Missing methods were included and other methods are now described in more detail.

3. All bar graphs should be modified to show individual data points so the reader can assess variability in the data.

We have now included the distribution of all data points for all bar graphs in the main figures and supplementary material.

Minor concerns:

4. The authors have not included reference to functional studies of multiple nearby residues that have been identified in patients with neurological disorders and are published (2A-N614S, 2A-N615K, 2B-N615I, 2B-N615K, 2B-N616K, see Strehlow et al., Li et al). I think this might be valuable for the reader.

In the last paragraph of the discussion we include now these important findings in patients with mutations in the NMDAR with respect to our analysis of the Grin2A gene targeted mice as requested.

In recent years numerous *de novo* GRIN2 variants have been identified in patients with neurological disorders. One set of rare mutations changes the activation profile of NMDARs and is located in the ligand-binding domain (LBD) of the NMDAR. A second set is located in the NMDAR channel pore of the channel gate⁹³. These mutations affect ion permeability and in particular the voltage-controlled Ca²⁺ influx. Carriers of these second set of mutations suffer from neurological dysfunction with different degrees of severity. Up to 500 rare disease variants in NMDAR genes have been identified in human patients. Most variants were found in GRIN2A and GRIN2B. The comparison of 249 individuals with pathogenic, or likely pathogenic, GRIN2A variants identified patients with severe developmental phenotypes associated with missense mutations in the ion pore or the linker domain. NMDAR mutations within the amino-terminal, ligand binding domain and null variants led to a less severe phenotype and were classified as 'loss of function mutations'^{94,95} whereas most of the severe NMDAR mutations with altered Mg²⁺ block were considered as gain of function mutations⁹⁵.

A detailed study with 12 *de novo* GRIN missense variants from 18 patients clearly showed that those missense mutations in the P2 loop of GluN1, GluN2A and GluN2B altered surface expression, pharmacological properties and other biophysical characteristics. It also demonstrated that these variants can have modest changes in agonist potency and proton inhibition. Furthermore, the voltage-dependent inhibition by Mg²⁺ was significantly reduced in all variants. Since the single channel conductance and Ca²⁺ permeability can be altered to different extents by different mutations, the degree of Ca²⁺ influx after glutamate stimulation is specific for each mutation. This means that the severity of any 'gain of function' mutation (like the *Grin2a*^(N615S)) and thus the severity of the associated phenotype will be defined by the kind of mutation⁹⁶. Li et al. summarize the phenotypes of 4 GluN2A(N614S), 2(N615K) and 1 GluN2A(N614I), 1(N615K) and 1(N616K) *de novo* mutations. Seven of these 9 patients showed muscle hypertonia, 6 suffered from epileptic seizures, and all exhibited intellectual disability and developmental delay. All GluN2A mutations were associated with language

problems and autism spectrum disorder was diagnosed in one GluN2A(N614S) patient and in one GluN2B(N615I) carrier. This is in line with other studies in heterozygous patients with the very same *de novo* NMDAR mutations who displayed a similar (although not identical) clinical manifestation^{93,95-97}.

Indeed, although two patients with the very same GluN2A(N614K) mutation had early-onset epileptic encephalopathies, there were also differences in their clinical manifestations which defied a clear genotype-phenotype correlation²⁷. Correspondingly, in our heterozygous *Grin2a*^{+/-S} mice the gain of function *Grin2a*^(N615S) mutation led - depending on the behavioral test and sometimes on the individual mouse - to either detectable or non-detectable phenotypes (for phenotypes that were clearly evident in the homozygous mutant mice). Since glutamate stimulation - and thus neuronal activity - is an obligatory trigger for Ca²⁺ influx through mutated NMDARs, these gain of function NMDAR mutations may be particularly sensitive to non-genetic-factors that affect neuronal activity.

5. Although the GluN2A-N615S variant is well characterized in terms of calcium permeation and magnesium block at different voltages, there are still multiple properties that have not been described in the literature. Agonist potency, deactivation, peak and steady-state amplitude, charge transfer, desensitization, efficacy of pore blockers (like memantine and MK-801 used in the study) and surface expression are unknown. Even though the aim of this paper was to perturb voltage-dependent block of GluN2A-containing NMDARs, changes in these other parameters might also impact neurophysiology. Thus, without an in vitro characterization of this mutation, it is difficult to conclude with certainty that this variant causes the described phenotypes due to loss of magnesium block alone. These are straightforward experiments, some of which should be added. Interestingly, the variant exists in human patients and some functional characterization has been performed, although apparently not published in the peer reviewed literature (<http://functionalvariants.emory.edu/database/index.html>). This kind of information would be a valuable addition and strengthen the overall conclusions.

We have now included the in vitro characterization of the GluN1/2(N615S) recombinant receptor in HEK293 cells as requested. Dr Pawlak, who performed the experiment is now included on the author list. We provide more information on the Mg²⁺ block, the desensitization kinetics in Fig. 1 b and some kinetic parameters in Supplementary Table 1. In the discussion we mention

Results:

Heterologous expression of GluN2A(N615S) (**Fig. 1a**) with GluN1 demonstrated a reduced Mg²⁺ block of GluN1/2(N615S) receptors in the presence of 1 and 4 mM of Mg²⁺ at hyperpolarized membrane potentials when compared with wild-type NMDARs (**Fig. 1b**). In the absence of Mg²⁺, short glutamate applications (20 ms) activated mutated and wild-type NMDAR channels with comparable current amplitudes and similar activation (rise time) and deactivation kinetics. During prolonged glutamate applications (600 ms) slower desensitization kinetics were obvious for the GluN1/2A(N615S) compared to GluN1/2A heterodimeric receptors (**Fig. 1b; Supplementary Table 1**).

Supplementary table 1 gives numbers for: Rise time/ms, τ des/ms * and τ decacw.

In addition we discussed the electrophysiological profile of recombinant GluN1/2A(M615S) receptors and the LTP in the first part of the discussion

6. It would be helpful to know whether the N-S mutation alters the IC50 or voltage dependence of memantine block. I wonder if actions of memantine reflect block of 2B subunits rather than 2A

mutant subunits, which should be insensitive to memantine. Some discussion of this possibility seems appropriate.

We are very thankful for this valuable comment. We agree with the reviewer that the effects of MK-801 and memantine most likely caused by a blockade of the GluN1/2B receptor. This is now mentioned in the results:

Since the NMDAR antagonists MK-801 and memantine reside within the channel vestibule, snuggling into the binding pocket,⁵⁴ and memantine as well as the MK801 blockade of NMDARs is reduced in recombinantly expressed NMDARs that carry mutations at analogous positions in the pore loop (e.g. GluN2A(N615K) GluN (N615Q) and GluN2B(N615Q))^{55,56}, it seems most likely that blockade of activity-responding GluN1/2B receptors is responsible for the genotype-specific MK801-induced c-Fos expression. The MK801-mediated inhibition of GluN1/2B activity of GABA-ergic interneurons has been proposed as a causal mechanism for the disinhibition of CA1 neurons in GluN2A deficient mice (Su et al., 2018).

7. The rationale behind the following conclusion is not well explained: “Together, these results show an incorporation of GluN2A(N615S) into synaptic NMDARs but the failure of GluN2A(N615S) receptors to participate in LTP.” Perhaps clarify or expand the text?

We expanded the text to clarify our conclusion removed the wrong statement “failure of GluN2A(N615S) receptors to participate in LTP”

To analyze whether these voltage-independent GluN1/2A(N615S) receptors can still induce synaptic plasticity, we analyzed field LTP (fLTP) at CA3-to-CA1 synapses in the different *Grin2a* genotypes. Here we found that the magnitude of hippocampal fLTP in *Grin2a^{S/S}* and *Grin2a^{+S}* mice was unaffected *ex vivo* and *in vivo* (**Fig. 2c, d**(Pawlak et al., 2005b)), in contrast to the reduced fLTP found in GluN2A-deficient mice and in mice lacking the GluN2A intracellular C-terminal domain of the GluN2A subunit (Sakimura et al., 1995; Sprengel et al., 1998). This suggests that the coincidence signaling of GluN1/2(N615S) receptors is still operative. However, since the GluN2B antagonist CP101,106 significantly reduced the fLTP in *Grin2a^{S/S}* and *Grin2a^{+S}* mice but not in *Grin2a^{+/+}* littermates (**Fig. 2e**), we conclude that (i) pure GluN1/2A(N615S) receptors have a reduced contribution to the long-term synaptic enhancement after tetanic stimulation, and (ii) the fLTP recorded in *Grin2a^{S/S}* and *Grin2a^{+S}* is substantially mediated by GluN2B-containing receptors.

This conclusion was strengthened by using four tetanic stimulations (4x 100 Hz), which can induce GluN2B-dependent LTP in the absence of functional GluN2A (Kiyama et al., 1998; Köhr et al., 2003). In comparison with single tetanic stimulation, LTP was significantly increased 40 – 45 min after the 4x 100 Hz stimulation in hippocampal slices of both *Grin2a^{S/S}* and *Grin2a^{+S}* mice compared to WT control littermates. This LTP increase was reduced by CP101,606 (**Supplementary Fig. 3a**), an effect that is reminiscent of the one described for LTP reduction in juvenile (P14) wild-type mice (Jensen et al., 2003) This LTP was still completely NMDAR-dependent and could be blocked by the NMDAR antagonist APV (**Supplementary Fig. 3b**). Together, these results show the incorporation of GluN2A(N615S) into synaptic NMDARs but reduced contribution of GluN2A(N615S) receptors in LTP.

8. The LTP work is well done, important for the manuscripts story, and should be part of figure 1.

We included part of the LTP work as Figure 2b-e and in the main text.

Fig. 2. Hippocampal synaptic transmission and plasticity in *Grin2a^{S/S}* and *Grin2a^{+S}* mice with the GluN2A(N615S) mutation

To directly assess changes in excitatory synaptic transmission and synaptic excitability, we recorded

simultaneously in the apical dendritic and soma layers in the CA1 region of hippocampal slices from GluN2A(N615S) expressing mice and wild-type littermates. First, we measured the fiber volley, the fEPSP and the population spike as a function of different stimulation strengths. In our field recordings the stimulation strength required to induce pre-volley amplitudes of 1.0 or 1.5 mV was statistically unaltered in GluN2A(N615S) expressing mice and showed only a trend towards lower fEPSP amplitudes at a given pre-volley amplitudes of 1.5 mV in *Grin2a^{S/S}* mice. Together with the similar paired-pulse ratio our field recordings revealed no major alterations in CA3-to-CA1 synaptic transmission of *Grin2a^{+/S}* and *Grin2a^{S/S}* mice (**Fig. 2b**).

To analyze whether these voltage-independent GluN1/2A(N615S) receptors can still induce synaptic plasticity, we analyzed field LTP (fLTP) at CA3-to-CA1 synapses in the different *Grin2a* genotypes. Here we found that the magnitude of hippocampal fLTP in *Grin2a^{S/S}* and *Grin2a^{+/S}* mice was unaffected *ex vivo* and *in vivo* (**Fig. 2c, d³¹**), in contrast to the reduced fLTP found in GluN2A-deficient mice and in mice lacking the GluN2A intracellular C-terminal domain of the GluN2A subunit^{32,33}. This suggests that the coincidence signaling of GluN1/2(N615S) receptors is still operative. However, since the GluN2B antagonist CP101,106 significantly reduced the fLTP in *Grin2a^{S/S}* and *Grin2a^{+/S}* mice but not in *Grin2a^{+/+}* littermates (**Fig. 2e**), we conclude that (i) pure GluN1/2A(N615S) receptors have a reduced contribution to the long-term synaptic enhancement after tetanic stimulation, and (ii) the fLTP recorded in *Grin2a^{S/S}* and *Grin2a^{+/S}* is substantially mediated by GluN2B-containing receptors.

This conclusion was strengthened by using four tetanic stimulations (4x 100 Hz), which can induce GluN2B-dependent LTP in the absence of functional GluN2A^{34,35}. In comparison with single tetanic stimulation, LTP was significantly increased 40 – 45 min after the 4x 100 Hz stimulation in hippocampal slices of both *Grin2a^{S/S}* and *Grin2a^{+/S}* mice compared to WT control littermates. This LTP increase was reduced by CP101,606 (**Supplementary Fig. 3a**), an effect that is reminiscent of the one described for LTP reduction in juvenile (P14) wild-type mice³⁶. This LTP was still completely NMDAR-dependent and could be blocked by the NMDAR antagonist APV (**Supplementary Fig. 3b**). Together, these results show the incorporation of GluN2A(N615S) into synaptic NMDARs but reduced contribution of GluN2A(N615S) receptors in LTP.

In the discussion the alterations of the electrophysiological profile with respect to the synaptic plasticity are outlined:

Our data obtained both on recombinant and on NMDARs from genetically modified animals shows that GluN2(N615S) containing NMDARs have remarkable reduced sensitivity to external Mg²⁺ ions as measured by the linearized I/V relation of GluN1/2A(N615S) channels and their less reduced voltage-controlled NMDARs currents in the presence of Mg²⁺ in HEK293 cells and hippocampal slices, respectively. Taking into account the 1.4 fold increased Ca²⁺ permeability (Wollmuth et al.,1996), no alterations in the channel conductance and proven surface and synaptic expression of GluN2(N615S)-containing NMDARs, all synaptic events are most likely associated with an increased NMDAR-mediated Ca²⁺ entry at resting membrane potential through synaptic channels. Moreover, spill-over of glutamate might produce additional Ca²⁺-influx through GluN1/2A(N615S) receptors into dendrites located close to active synapses and will amplify the impact of volume transmission to the network activity and the disturbance of synaptic plasticity in GluN2A(N615S) expressing mice.

Alteration in synaptic plasticity could be monitored by an increased LTP component of GluN2B containing NMDARs in hippocampi from GluN2A(N615S) expressing mice. Very similar to our findings of an increased GluN2B component at CA3-toCA1 synapses in young mice³⁵, the LTP could be significantly enhanced by repeated tetanic stimulations and the enhancement could be blocked by CP101,106 in adult *Grin2a^{+/S}* and *Grin2a^{S/S}* mice. This might indicate that the altered Ca²⁺ homeostasis by the glutamate-triggered Ca²⁺ influx through the mutant NMDARs keeps the plasticity mechanism of CA1 cell in an immature state in GluN2A(N615S) expressing mice.

9. *The overall layout of the figures does not transparently illustrate the data or variability. For example, figure 1 shows representative bands for a western blot but no density measurements. Inside the figure legend, it states “t-test, p>0.05”. Where is the densitometry data, how many observations were made, etc. This issue is repeated in other figures.*

In Fig. 1C the quantification of the immunoblots is given as bar graphs with the distribution for all three genotypes. The immunoblots used for the quantification are all presented in the Supplementary Fig. 2.

Supplementary Fig. 2: Expression levels of reference proteins in GluN2A(N615S) expressing mice

10. *Figure 2 legend doesn't mention whether quantification or any statistical test was performed. AS written, it appears that this is a result of a single experiment, although I am certain the authors replicated the data an appropriate number of times. Perhaps expand discussion to indicate number of experiments performed, quantification, etc..*

The number of animals used for Fig.3 (previously Fig.2) are now mentioned in the legend to the figure: Number of animals is given below the bars. For the Nissl stain, Tunnel test and Timm stain 3 animals were used per genotype. For the immunohistological analysis of GFAP, NeuN, CB and PV 5 animals and for the GluA1 IHF stain 3 animals were used.

11. *The GluA1 representative IHC images in Fig 2 look overexposed to a point where differences in GluA1 expression levels might be missed.*

We apologize for the poor quality of the GluA1 IHC. We now show a picture with a higher resolution and improved contrast. The different hippocampal layers are now more visible and comparable.

12. *Manuscript has a lot of data that could be organized more clearly. Perhaps consider combining Figures 6 and 7 into one figure, with some portions going to supplemental? This is only a suggestion, and the authors should simply consider the best way to organize data.*

We have shortened the section of the main manuscript on learning and behavior, and have moved the detailed experimental protocols of the T-mazes, Y-maze and Water-mazes to the Methods section (while also providing additional details as requested). Furthermore the confirmatory but independent behavioral experiments in Figs. 6 and 7 are now moved to Supplementary Fig. 7 and only the key learning experiments are now given in one main Figure (Fig. 8).

13. *Typo in abstract, accidentally say ‘NMDR’ instead of ‘NMDAR’*

“NMDR” is now changed to “**NMDAR**”

14. *Typo in abstract and in introduction where “GLUN2A” is in all capital letters*

“GLUN2A” is now changed to **GluN2A**

REVIEWERS' COMMENTS:

Reviewer #1 (Remarks to the Author):

The authors have addressed all my concerns and have done a great job at improving the description of methods and strengthening the manuscript.

Reviewer #2 (Remarks to the Author):

The authors have addressed my concerns.

Reviewer #3 (Remarks to the Author):

The authors have addressed all of my criticisms. Congratulations on an excellent and informative study.